# Mitochondrial Ca²⁺ and membrane potential, an alternative pathway for Interleukin 6 to regulate CD4 cell effector function

Rui Yang[1†], Dario Lirussi[1†], Tina M Thornton[1], Dawn M Jelley-Gibbs[2‡], Sean A Diehl[1], Laure K Case[1], Muniswamy Madesh[3,4], Douglas J Taatjes[5,6], Cory Teuscher[1], Laura Haynes[2§], Mercedes Rincón[1*]

[1]Department of Medicine, Immunobiology Program, University of Vermont, Burlington, United States; [2]Trudeau Institute, Saranac Lake, United States; [3]Department of Medical Genetics and Molecular Biochemistry, Temple University, Philadelphia, United States; [4]Center for Translational Medicine, Temple University, Philadelphia, United States; [5]Department of Pathology and Laboratory Medicine, University of Vermont, Burlington, United States; [6]Microscopy Imaging Center, University of Vermont, Burlington, United States

*For correspondence: mrincon@uvm.edu

Present address: †Department of Vaccinology and Applied Microbiology, Helmholtz Center for Infection Research, Braunschweig, Germany; ‡Taconic, Germantown, United States; §Center on Aging and Department of Immunology, University of Connecticut Health Center, Farmington, United States

Competing interests: The authors declare that no competing interests exist.

**Abstract** IL-6 plays an important role in determining the fate of effector CD4 cells and the cytokines that these cells produce. Here we identify a novel molecular mechanism by which IL-6 regulates CD4 cell effector function. We show that IL-6-dependent signal facilitates the formation of mitochondrial respiratory chain supercomplexes to sustain high mitochondrial membrane potential late during activation of CD4 cells. Mitochondrial hyperpolarization caused by IL-6 is uncoupled from the production of ATP by oxidative phosphorylation. However, it is a mechanism to raise the levels of mitochondrial Ca²⁺ late during activation of CD4 cells. Increased levels of mitochondrial Ca²⁺ in the presence of IL-6 are used to prolong *Il4* and *Il21* expression in effector CD4 cells. Thus, the effect of IL-6 on mitochondrial membrane potential and mitochondrial Ca²⁺ is an alternative pathway by which IL-6 regulates effector function of CD4 cells and it could contribute to the pathogenesis of inflammatory diseases.

## Introduction

Interleukin 6 (IL-6) is an inflammatory cytokine that is elevated in several autoimmune and inflammatory disorders, including rheumatoid arthritis (RA) (*Kishimoto, 2005*). Inhibition of IL-6 signaling by an anti-IL-6R antibody has been shown to be a highly effective therapy in treating patients with RA (*Tanaka and Kishimoto, 2012*). IL-6 plays crucial role in regulating CD4 T helper cell differentiation and cytokine production (*Dienz and Rincon, 2009*). It enhances Th2 differentiation through an auto-feedback loop by upregulating autocrine IL-4 production (*Rincon et al., 1997*; *Diehl et al., 2002*). IL-6 inhibits IFNγ production and Th1 differentiation (*Diehl et al., 2000*). In combination with TGFβ, IL-6 also contributes to the differentiation of Th17 cells (*Bettelli et al., 2006*; *Ivanov et al., 2006*; *Zhou et al., 2007*). IL-6 inhibits regulatory T cell function and downregulates Foxp3 expression (*Pasare and Medzhitov, 2003*; *Dienz and Rincon, 2009*). In addition, IL-6 alone, without the need of TGFβ, induces IL-21 expression, a mechanism by which it promotes the generation of follicular T helper (Tfh) cells (*Nurieva et al., 2008*; *Suto et al., 2008*; *Dienz et al., 2009*; *Diehl et al., 2012*).

**eLife digest** Inflammation is a normal part of the body's response to an infection or injury and it helps to start the healing process. However, if left unchecked, inflammation itself can damage tissues, and diseases such as rheumatoid arthritis are the result of uncontrolled inflammation.

Certain immune cells release molecules that can either trigger or suppress inflammation. Interleukin 6 is an example of a 'pro-inflammatory' molecule, which regulates the activity of groups of immune cells collectively known as 'CD4 cells'. People who are overweight or obese have higher levels of interleukin 6 than people of a healthy weight. Obesity and other metabolic conditions have been linked to problems with structures called mitochondria, which make a molecule called ATP that provides cells with the energy they need to survive. But it is not known if interleukin 6 can affect the activity of mitochondria inside CD4 cells.

Now, Yang et al. have discovered that interleukin 6 can affect the mitochondria inside CD4 cells and, in doing so, have identified a new way that interleukin 6 can regulate these cells' activity. Experiments involving immune cells from mice revealed that interleukin 6 triggers a cascade of signaling events that aid the formation of so-called 'mitochondrial respiratory chain supercomplexes' in CD4 cells. These are groups of proteins that work together in the membranes of mitochondria and are vital for the activity of these structures. The formation of these supercomplexes maintains a large voltage difference across the membrane of the mitochondria that occurs during the later stages of CD4 cell activation.

Yang et al. found that this voltage difference was not linked to the production of ATP, but that it did raise the levels of calcium ions inside the mitochondria. Further experiments revealed that these increased levels of calcium ions prolong the production of other pro-inflammatory molecules in the CD4 cells.

Following the discovery of a new pathway that regulates the activity of CD4 cells, the next challenge is to see if the parts of this pathway could be targeted with drugs to help treat inflammatory diseases such as rheumatoid arthritis. Moreover, because interleukin 6 plays an active role in other diseases such as cancer, further studies of this new pathway may help explain how this molecule encourages cancers to progress and/or spread around the body.

---

IL-6 binds to its membrane receptor, which triggers signaling through gp130, a common transducer that activates Jak/Stat3 and Ras/MAPK pathways in T cells (*Boulanger et al., 2003*; *Heinrich et al., 2003*; *Kishimoto, 2005*). Stat3 is a transcription factor present in cytosol but translocates to the nucleus upon stimulation where it mediates the expression of numerous genes. Stat3 has been previously implicated in the regulation of genes involved in cell survival and proliferation by directly binding to multiple survival genes, including *Bcl2*, *Fos*, *Jun*, *Mcl1* and *Fosl2* (*Hirano et al., 2000*; *Bourillot et al., 2009*; *Durant et al., 2010*; *Carpenter and Lo, 2014*). Additionally, IL-6-dependent Stat3 activation plays an important role in the expression of several cytokine genes, including *Il21* and *Il17* (*Mathur et al., 2007*; *Zhou et al., 2007*; *Dienz et al., 2009*). In addition to its role as a nuclear transcription factor, Stat3 has been found within mitochondria in liver, heart and some cell lines where it enhances the mitochondrial respiratory chain activity (*Gough et al., 2009*; *Wegrzyn et al., 2009*). However, no studies have addressed whether IL-6 regulates mitochondrial function through Stat3.

IL-6 has for long been associated with metabolic changes and high levels of IL-6 in serum have been correlated with BMI (*Mohamed-Ali et al., 1997*; *Fried et al., 1998*; *Vgontzas et al., 2000*). Recent studies indicate that IL-6 is linked to glucose homeostasis in adipose tissue and it participates in the switch from white to brown fat tissue in cancer-induced cachexia (*Stanford et al., 2013*; *Petruzzelli et al., 2014*). However, it remains unclear whether IL-6 has a direct effect on the metabolism of cells. But in the context of ischemia-reperfusion injury in cardiomyocytes, IL-6 has been shown to maintain mitochondrial membrane potential (MMP) in cardiomyocytes (*Smart et al., 2006*). Despite the known role of IL-6 in the CD4 cell effector function, no studies have addressed whether IL-6 has an effect on mitochondrial function in CD4 cells.

Here we show that IL-6 plays an important role in maintaining MMP late during CD4 cell activation in a Stat3-dependent manner. IL-6-mediated mitochondrial hyperpolarization is, however, uncoupled

from the oxidative phosphorylation and ATP production. Instead, IL-6 uses the high MMP to raise mitochondrial $Ca^{2+}$ and, consequently, cytosolic $Ca^{2+}$ levels to promote cytokine expression late during activation. Thus we have identified a previously undescribed mechanism by which IL-6 regulates CD4 cell effector function.

## Results

### IL-6 is essential to sustain MMP during activation of CD4 cells

Although the role of IL-6 in CD4 cell differentiation and cytokine gene expression is well established, little is known about the role of this cytokine in mitochondrial function. An essential function of the mitochondrial electron transport chain (ETC), in addition to the transfer of electrons, is the generation of an electrochemical gradient across the mitochondrial inner membrane by accumulating $H^+$ at the intermembrane space. This electrochemical gradient, known as MMP, is used as a mechanism to generate ATP. Since IL-6 has been associated with maintaining MMP in cardiomyocytes (*Smart et al., 2006*), we examined whether IL-6 regulates the MMP in CD4 cells during activation. Fresh CD4 cells were activated with anti-CD3 and anti-CD28 antibodies (Abs) in the presence or absence of IL-6 for different periods of times, stained with TMRE (an MMP indicator), and analyzed by flow cytometry. Most freshly isolated CD4 cells were hyperpolarized as shown by the high TMRE staining (*Figure 1A*). However, cells activated in the absence of IL-6 depolarized progressively during activation (*Figure 1A*). Interestingly, the presence of IL-6 prevents mitochondrial depolarization during CD4 cell activation (*Figure 1A*). After 48hr of activation, most CD4 cells activated in the presence of IL-6 maintained a high MMP (TMRE^high) (*Figure 1B*). In contrast to IL-6, the presence of exogenous IL-2, the main growth factor of T cells, did not affect MMP in activated CD4 cells (*Figure 1C*), supporting a selective role for IL-6 on MMP.

To examine the effect of IL-6 on mitochondrial mass and levels of ETC complexes, we performed Western blot analysis for subunits of these complexes using whole cell extracts. IL-6 did not affect the overall mitochondrial mass as determined by the levels of COX IV (Complex IV subunit of ETC), NDUFS3 and NDUFA9 (Complex I subunits) (*Figure 1D*). In addition, the frequency of live cells among those activated in the presence of IL-6 was not significantly different from the frequency of live cells in the absence of IL-6 (*Figure 1E*). Thus, the increase of MMP triggered by IL-6 is not a consequence of survival or change in mitochondrial mass.

Antigen presenting cells (APCs) are one of the major sources of IL-6 during CD4 cell activation. To examine whether IL-6 was required to maintain the mitochondrial hyperpolarization during antigen activation, naive CD4 cells were obtained from OT-II TCR transgenic mice (*Barnden et al., 1998*) and activated with OVA peptide and APCs isolated from WT or IL-6 KO mice. Similar to CD4 cells activated with anti-CD3/CD28 Abs in the presence of IL-6, a large frequency of OT-II CD4 cells activated with WT APC showed a high MMP (*Figure 1F,G*). However, a blocking anti-IL-6 Ab drastically decreased the frequency of cells with high MMP (*Figure 1F,G*). In contrast to WT APCs, very low frequency of activated CD4 cells showed high MMP when APC from IL-6 KO mice were used (*Figure 1F,G*). Remarkably, addition of exogenous IL-6 to cells activated with IL-6 KO APCs restored high MMP (*Figure 1F,G*). Thus, these results indicate that IL-6 derived from APC during in vitro activation of CD4 cells is essential to maintain mitochondrial hyperpolarization.

To address the role of IL-6 in regulating the MMP in CD4 cells during in vivo activation, we performed adoptive transfer of OT-II CD4 cells into WT or IL-6 KO mice as hosts. Mice were then immunized with ovalbumin, and after two days, cells were harvested to examine their MMP. Similar to in vitro results, the fraction of OT-II cells maintaining a high MMP was significantly greater in WT mice relative to IL-6 KO mice (*Figure 1H*). Together, these results indicate that IL-6 plays an essential role in maintaining the MMP during activation of CD4 cells.

### IL-6 facilitates the formation of respiratory chain supercomplexes in CD4 cells during activation

Morphological states of highly pleomorphic inner membrane cristae reflect the different mitochondrial metabolic stages. Mitochondrial cristae shape has been shown to influence the efficiency of the respiratory chain in part by affecting the formation of respiratory chain supercomplexes (RCS) (*Hackenbrock, 1966*; *Gomes et al., 2011*; *Cogliati et al., 2013*), formed of Complex I together with Complex III and Complex IV. The function of RCS is to facilitate the transfer of electrons between

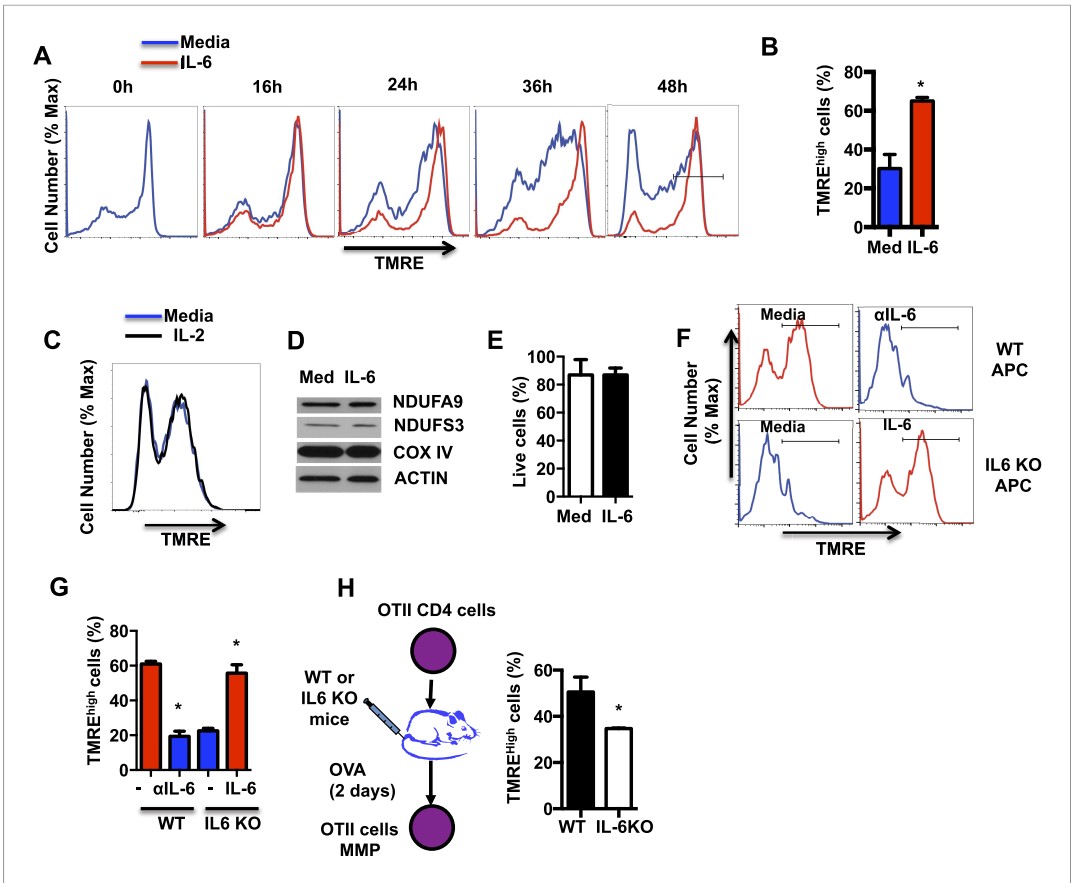

**Figure 1**. IL-6 sustains high mitochondrial membrane potential (MMP) late during activation. (**A**) MMP during activation of CD4 cells with anti-CD3/CD28 Abs over time in the presence or absence of IL-6, as determined by staining with TMRE and flow cytometry analysis. (**B**) Percentage of CD4 cells with TMRE[high] (defined by the gate displayed in (**A**) at 48 hr, after activation as in (**A**) (n = 3). (**C**) MMP during activation of CD4 cells in the absence or presence of IL-2 was determined by staining with TMRE and flow cytometry analysis. (**D**) Expression of NDUFA9, NDUFS3, COX IV and ACTIN examined by Western blot analysis using whole-cell extracts from CD4 cells activated for 48 hr. (**E**) Percentage of live CD4 cells activated as in (**A**) for 48 hr, determined by flow cytometry. (n = 3). (**F**) MMP in OT-II CD4 cells activated by WT or IL-6 KO APCs with OVA peptide in the presence or absence of the supplement of exogenous of IL-6 (IL-6) or blocking anti-IL-6 antibody (αIL-6) for 48 hr. (n = 3). (**G**) Percentage of TMRE[high] population in OT-II CD4 cells from (**F**) (n = 3). (**H**) OT-II CD4 cells were adoptively transferred to WT or IL-6 KO recipient mice that were then immunized with ovalbumin (and Alum). After 2 days, cells were harvested to examine for MMP. Percentage of TMRE[high] population in activated OT-II T cells from WT or IL-6 KO mice were determined by TMRE staining and flow cytometry analysis. Error bars represent the mean ± SD. *denotes p < 0.05, as determined by Student's t test. Results are representative of 2–3 experiments.

complexes and increase Complex I activity while reducing the electron leak from ETC and mitigate the production of reactive oxygen species (ROS) (*Schägger, 1995*; *Acín-Pérez et al., 2008*; *Althoff et al., 2011*; *Winge, 2012*). To determine whether IL-6 could affect cristae shape, we examined CD4 cells activated in the presence or absence of IL-6 by transmission electron microscopy (TEM) imaging. No obvious differences in mitochondrial integrity or mitochondrial mass were observed in cells activated with or without IL-6 (*Figure 2A*). Similarly, there was no increase in the number of mitochondria in CD4 cells activated in the presence of IL-6 (*Figure 2—figure supplement 1*). However, the morphology of the mitochondrial cristae in cells activated with IL-6 was different from that of cells activated without IL-6 (*Figure 2A*). The number of mitochondria with expanded and disorganized cristae was greater in CD4 cells activated in the absence of IL-6 compared with CD4 cells activated with IL-6 (*Figure 2B*). In contrast, the number of mitochondria with tight and organized cristae was higher in cells activated

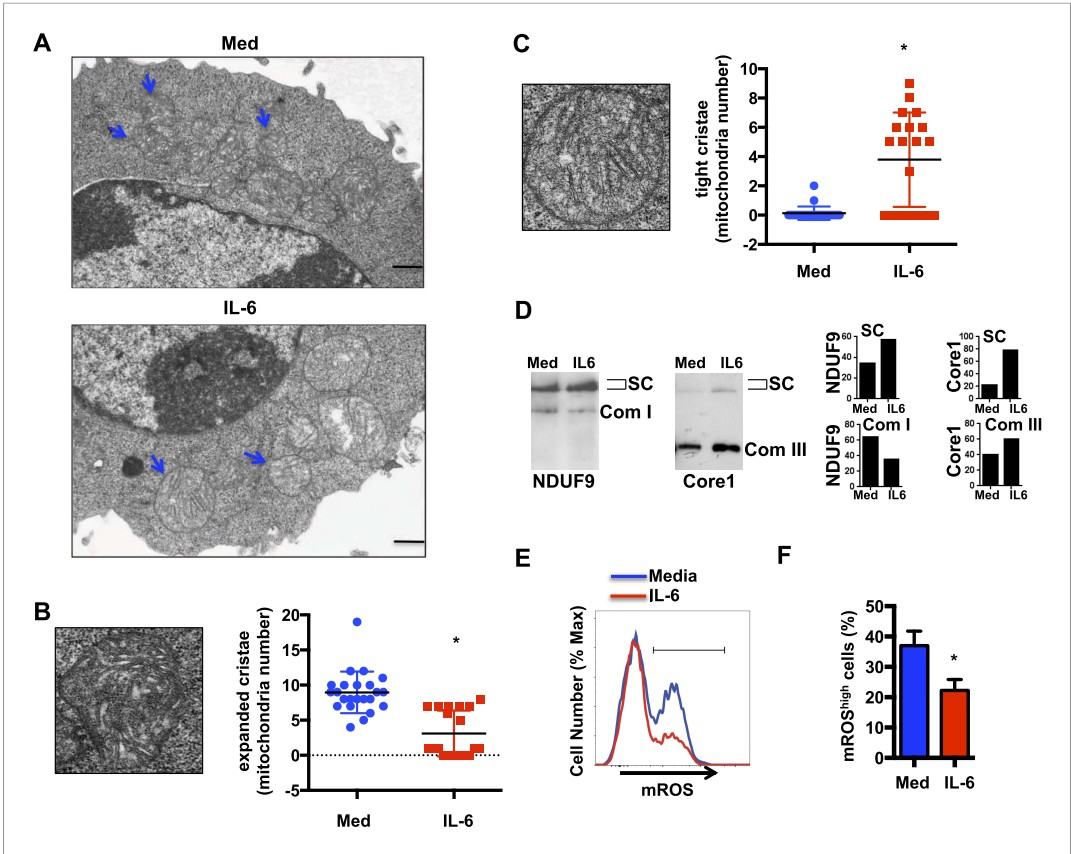

**Figure 2**. IL-6 facilitates the formation of respiratory chain supercomplexes in CD4 cells during activation. (**A**) Transmission electron microscopy analysis of mitochondria in CD4 cells activated in the presence or absence of IL-6. Original magnification, 6,000×. Bars represent 500 nm; Blue arrows indicate mitochondria (**B**) Representative image of an 'expanded cristae' mitochondrion (Left). Number of mitochondria with expanded cristae in CD4 cells activated in the presence or absence of IL-6 (right). (n = 25). (**C**) Representative image of a 'tight cristae' mitochondrion (Left). Number of mitochondria with tight cristae in CD4 cells activated in the presence or absence of IL-6 (right). (n = 23). Error bars represent the mean ± SD. *denotes $p < 0.05$, as determined by Student's t test. Results are representative of 2 experiments. (**D**) Digitonin-soluble mitochondrial extracts from CD4 cells were resolved by BN-PAGE and transferred onto a membrane (Western blot) and immunoblotted for NDUFA9 and Core1 protein. Immunoreactivity for the two proteins within the supercomplex (SC) region is shown. Immunoreactivity for NDUFA9 with monomeric complex I (Com I) and Core1 with dimeric complex III (Com III) are shown. Lower panels display the densitometry of NDUFA9 (left) and Core I (right) subunits within the supercomplex region (SC) and densitometry at the individual Complex I (Com I) and Complex III (Com III) respectively. (**E**) Mitochondrial ROS during activation of CD4 cells with anti-CD3/28 Abs in the presence or absence of IL-6 for 48 hr was determined by staining with MitoSox and flow cytometry analysis. (**F**) Percentage of CD4 cells with mROS^high, defined by the gate displayed in (**E**) at 48 hr, after activation as in (**E**) (n = 4). Error bars represent the mean ± SD. *denotes $p < 0.05$, as determined by Student's t-test. Results are representative of 2 experiments.

The following figure supplement is available for figure 2:

**Figure supplement 1**. IL-6 does not increase the number of mitochondria in CD4 cells during activation.

in the presence of IL-6 (*Figure 2C*). Thus, IL-6 affects mitochondrial cristae shape during activation of CD4 cells.

To determine whether the effect of IL-6 on the mitochondrial cristae morphology could be reflected in an altered formation of RCS as a mechanism to maintain a high MMP, we examined the presence of RCS in activated CD4 cells. We performed blue-native gel electrophoresis (BN-PAGE) using mitochondrial extracts generated in the presence of digitonin to preserve the supercomplexes

(SCs) (*Acín-Pérez et al., 2008*), followed by Western blot analysis. The levels of RCS but not the levels of individual Complex I or Complex III were increased in mitochondria from IL-6-stimulated CD4 cells, as determined by the presence of NDUFA9 (Complex I) and Core I (Complex III) within the RCS region (*Figure 2D*).

Since the formation of RCS is associated with increased MMP but reduced mitochondrial ROS (mROS) (*Schägger, 1995*; *Acín-Pérez et al., 2008*; *Althoff et al., 2011*; *Winge, 2012*), we examined the production of mROS in CD4 cells activated with or without IL-6 by flow cytometry analysis using MitoSOX, a mitochondrial superoxide indicator. Despite the increased MMP, IL-6 reduced the production of mROS (*Figure 2E,F*). Thus, the formation of RCS facilitated by IL-6 makes possible for this cytokine to sustain mitochondria hyperpolarization while minimizing the production of mROS during activation of CD4 cells.

## IL-6-mediated mitochondrial hyperpolarization is uncoupled from OXPHOS

The energy released from the transport of $H^+$ from the mitochondrial intermembrane space to the mitochondrial matrix through $F_0F_1$ ATP synthase, a subunit of Complex V, is coupled to ATP generation. Thus, an increased MMP elicited by IL-6 could potentially lead to an increase in mitochondrial ATP synthesis. We therefore examined ATP production in CD4 cells activated in the presence or absence of IL-6. Surprisingly, despite of the increased MMP, intracellular ATP levels were not affected by IL-6 (*Figure 3A*). TCR stimulation has been shown to trigger rapid ATP release from CD4 cells (*Yip et al., 2009*). It was therefore possible that IL-6 increased ATP synthesis but also ATP release. However, analysis of ATP levels in culture supernatants of activated cells showed no difference in the levels of extracellular ATP (*Figure 3B*). Since most ATP in activated T cells is generated through glycolysis (*Pearce et al., 2013*), increased MMP by IL-6 could enhance mitochondrial oxidative phosphorylation (OXPHOS) but have minimal effect on overall ATP levels. To further address the effect of IL-6 on mitochondrial OXPHOS, we examined oxygen consumption rate (OCR) using the Seahorse XF24 analyzer (*Wu et al., 2007*). No statistically significant difference in basal mitochondrial OCR or maximal respiratory capacity was detected (*Figure 3C*). Thus, the effects of IL-6 on the MMP are uncoupled from OXPHOS.

We also examined whether the mitochondrial hyperpolarization by IL-6 could compromise anaerobic glycolysis during activation. Culture supernatants of activated CD4 cells with or without IL-6 were assayed for lactate production. Lactate production was not significantly different in cells activated with IL-6 (*Figure 3D*). The Seahorse XF24 analyzer was also used to measure the extracellular acidification rate (ECAR), another alternative approach to examine the rate of glycolysis. Consistent with the production of lactate, there was no difference in anaerobic glycolysis in the presence of IL-6 during CD4 cell activation (*Figure 3E*). Thus, although IL-6 maintains high MMP late during the activation of CD4 cells, it does not alter rates of OXPHOS or anaerobic glycolysis.

## IL-6-mediated high MMP results in elevated mitochondrial Ca$^{2+}$ levels

Although the main function of the MMP is to drive the generation of ATP through OXPHOS, MMP also plays an important role in mitochondrial Ca$^{2+}$ homeostasis (*Rizzuto et al., 2012*). Mitochondria are emerging as the primary subcellular Ca$^{2+}$ store which buffers cytosolic Ca$^{2+}$ (*Starkov, 2010*). Mitochondrial Ca$^{2+}$ uptake is modulated by mitochondrial calcium uniporter (MCU) and it is dictated by the MMP (*Baughman et al., 2011*; *De Stefani et al., 2011*; *Mallilankaraman et al., 2012a*, *2012b*; *Shanmughapriya et al., 2015*), while Ca$^{2+}$ release from mitochondria is mediated by the mitochondrial Na$^+$/Ca$^{2+}$ exchanger (mNCLX) (*Kirichok et al., 2004*; *Palty et al., 2010*; *Nita et al., 2012*; *Rizzuto et al., 2012*). Upon TCR engaging, it has been reported that the formation of the immunological synapse triggers early store-dependent Ca$^{2+}$ influx through mitochondrial Ca$^{2+}$ buffering (*Hoth et al., 1997*; *Quintana et al., 2007*). However, little is known about the mitochondrial Ca$^{2+}$ signaling in activated effector cells and how it may contribute to CD4 cell effector functions. We examined whether an increased MMP regulated by IL-6 could affect mitochondrial Ca$^{2+}$ homeostasis. CD4 cells activated in the presence or absence of IL-6 for 48 hr were stained with Rhod-2 AM, a selective indicator for mitochondrial Ca$^{2+}$ (*Hajnoczky et al., 1995*; *Brisac et al., 2010*) and analyzed by flow cytometry. Consistent with an increased MMP, there was a significantly greater frequency of cells with high levels of mitochondrial Ca$^{2+}$ (Rhod-2$^{high}$) in the presence of IL-6 (*Figure 4A,C*). Short treatment of IL-6-activated CD4 cells with the depolarizing agent Carbonyl cyanide m-chlorophenyl hydrazone (CCCP) significantly reduced the

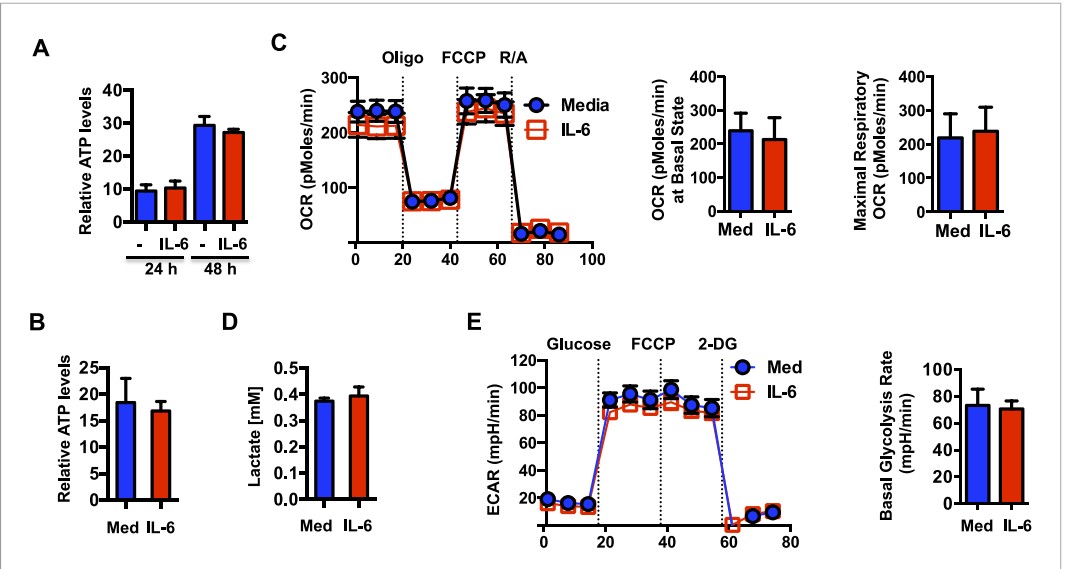

**Figure 3**. IL-6-mediated increase in mitochondrial membrane potential in CD4 cells is uncoupled from OXPHOS. (**A**) Intracellular ATP levels (per $10^4$ cells) in CD4 cells activated in the presence (IL-6) of absence of IL-6 (Med) (n = 5). (**B**) Extracellular ATP levels in supernatants of CD4 cells activated for 48 hr (n = 3). (**C**) Oxygen consumption rates in CD4 cells activated with or without IL-6 for 48 hr, under basal conditions and in response to oligomycin (oligo), FCCP or rotenone plus antimycin (R/A). Average of basal level OCR (n = 3) and the average of maximal OCR (n = 3) are shown. (**D**) Lactate levels in supernatant of CD4 cells activated for 48 hr (n = 3). (**E**) Extracellular acidification rates (ECAR) were measured in activated CD4 cells (48 hr) under basal conditions or in response to glucose, FCCP or 2-deoxyglucose (2-DG) sequentially. Average of basal ECAR levels are graphed on the right (n = 3). Error bars represent mean ± SD. No statistically significant differences (p > 0.05) were found for any of the assays, as determined by Student's t-test or two-way ANOVA. Results are representative of 2–3 experiments.

frequency of cells with high levels of mitochondrial $Ca^{2+}$ (Rhod-2[high]) (*Figure 4B,C*), indicating that the increased levels of mitochondrial $Ca^{2+}$ are dependent on mitochondrial hyperpolarization.

Because of their dynamic characteristics and ability to redistribute within the cell, mitochondria play an important role in cytoplasmic $Ca^{2+}$ homeostasis. Mitochondria uptake $Ca^{2+}$ through MCU at the cytoplasmic membrane near the extracellular calcium channels, as well as from ER storage, and serve as a delivery vehicle to increase cytosolic $Ca^{2+}$ (*Rizzuto et al., 2012*; *Soboloff et al., 2012*). Thus, early during T cell activation mitochondria have been shown to relocate close to immune synapse and contribute to increase cytosolic $Ca^{2+}$ (*Quintana et al., 2007*; *Schwindling et al., 2010*). To determine whether the increase in mitochondrial $Ca^{2+}$ elicited by IL-6 could affect the levels of free cytosolic $Ca^{2+}$, we examined the basal level of cytosolic $Ca^{2+}$ in CD4 cells using Fura-2 AM as a calcium indicator. The levels of cytosolic $Ca^{2+}$, as determined by fluorometric ratio at 340 nm/380 nm ($F_{340/380}$), in cells activated with IL-6 were higher than in cells activated without IL-6 (*Figure 4D*). It has been previously shown that TCR stimulation fails to induce cytosolic $Ca^{2+}$ flux in activated CD4 cells, as determined by flow cytometry analysis (*Nagaleekar et al., 2008*). Similarly, no $Ca^{2+}$ flux was triggered by TCR stimulation in CD4 cells activated in the presence of IL-6 (data not shown). However, similar to the results with Fura-2 staining, analysis of the cytosolic $Ca^{2+}$ baseline by Indo-1 staining and flow cytometry analysis also revealed higher baseline in CD4 cells activated in the presence of IL-6 relative to cells activated in the absence of IL-6 (*Figure 4—figure supplement 1A*). Maximum cytosolic $Ca^{2+}$ levels triggered by the calcium ionophore, ionomycin were comparable between CD4 cells activated in the presence or absence of IL-6 (*Figure 4—figure supplement 1A*). Thus, the presence of IL-6 during activation maintains increased levels of cytosolic $Ca^{2+}$.

To demonstrate that this increased cytosolic $Ca^{2+}$ was dependent on high mitochondrial $Ca^{2+}$, we examined the effect of CGP-37157, a blocker of mitochondrial $Ca^{2+}$ efflux (*Cox et al., 1993*). As previously demonstrated (*Delmotte et al., 2012*), treatment with CGP-37157 resulted in increased levels of mitochondrial $Ca^{2+}$ (*Figure 4—figure supplement 1B*). Importantly, the treatment with

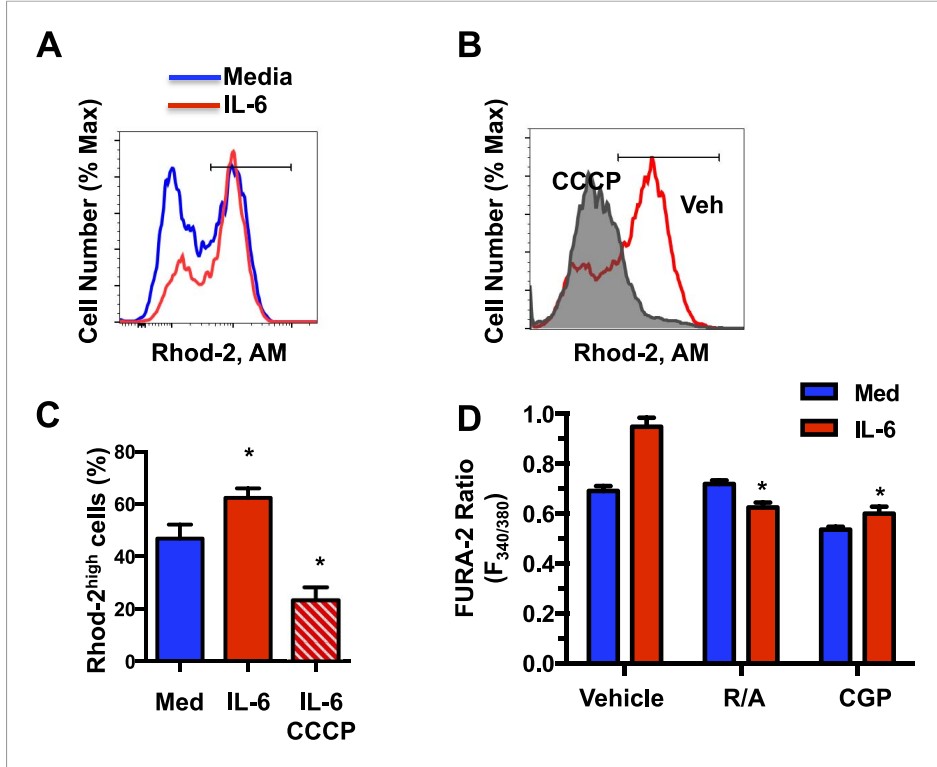

**Figure 4**. IL-6-mediated high MMP results in elevated mitochondrial $Ca^{2+}$ levels. (**A**) Mitochondrial $Ca^{2+}$ in CD4 cells activated in absence or presence of IL-6 for 48 hr was determined by staining with Rhod-2 AM and flow cytometry analysis. (**B**) Rhod-2 staining in CD4 cells activated with IL-6 for 46 hr followed by incubation with CCCP or vehicle (Veh) for 2 hr. (**C**) Percentage of Rhod-2$^{high}$ population in CD4 cells activated as in (**A**). Gates are shown in (**A**) and (**B**) (n = 4). (**D**) CD4 cells were activated for 48 hr with or without IL-6 and treated for the last 4 hr with vehicle, rotenone plus antimycin (R/A) or CGP-37157 (CGP). Cytoplasmic $Ca^{2+}$ was measured using Fura-2 AM staining. Fluorometric ratio at 340 nm/380 nm ($F_{340/380}$) is shown. (n = 3). Error bars represent the mean $\pm$ SD. *denotes $p < 0.05$, as determined by Student's t test and one-way or two-way ANOVA test. Results are representative of 2–3 experiments. DOI: 10.7554/eLife.06376.007

The following figure supplement is available for figure 4:

**Figure supplement 1**. IL-6 maintains elevated cytosolic $Ca^{2+}$ through its effect on the MMP and mitochondrial $Ca^{2+}$. DOI: 10.7554/eLife.06376.008

CGP-37157, lowered the cytosolic $Ca^{2+}$ levels in CD4 cells activated in the presence of IL-6 to the levels found in those without IL-6 (*Figure 4D*), indicating that this increase was dependent on mitochondrial $Ca^{2+}$. In addition, reducing the MMP in IL-6-stimulated cells by treatment with inhibitors of Complex I (rotenone) and Complex III (antimycin) also lowered the levels of cytosolic $Ca^{2+}$ (*Figure 4D*). Similar effects were found by the treatment with CCCP (*Figure 4—figure supplement 1C*). IL-6 therefore provides a mechanism for CD4 cells to maintain elevated levels of cytosolic $Ca^{2+}$ through its effect on the MMP and mitochondrial $Ca^{2+}$.

## The regulation of the MMP and mitochondrial $Ca^{2+}$ elicited by IL-6 is Stat3 dependent

In addition to its role as a transcription factor, several studies have shown the presence of Stat3 in mitochondria where it regulates the ETC primarily in tissues with high mitochondria content (*Gough et al., 2009*; *Wegrzyn et al., 2009*; *Heusch et al., 2011*; *Lachance et al., 2013*; *Zhang et al., 2013*; *Erlich et al., 2014*). Although IL-6 is a major activator of Stat3, no studies have previous address the regulation of Stat3 in mitochondria by this cytokines. However, the maintenance of high MMP late during activation of CD4 cells by IL-6 could possibly be mediated by Stat3. We first examined whether Stat3 could also be present in mitochondria in activated CD4 cells by Western blot analysis using

extracts from different subcellular fractions. As expected, Stat3 was present in both the nucleus and cytosol (*Figure 5A*). Interestingly, however, high levels of Stat3 were also present in mitochondria (*Figure 5A*). GAPDH and COX IV were used as cytosolic and mitochondrial fraction markers, respectively (*Figure 5A*). To examine whether localization of Stat3 in mitochondria was influenced by IL-6 during CD4 cell activation, we performed Western blot analysis using mitochondrial extracts from CD4 cells activated in the presence or absence of IL-6 as well as from freshly isolated CD4 cells.

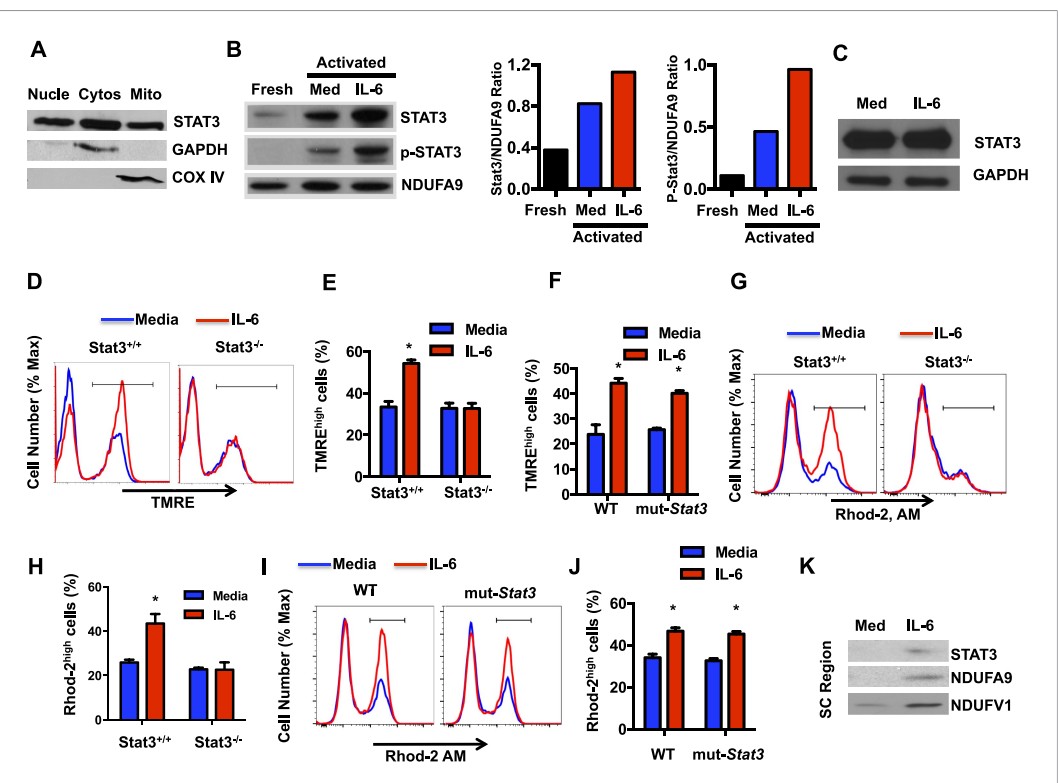

**Figure 5**. The regulation of the MMP and mitochondrial $Ca^{2+}$ elicited by IL-6 is Stat3 dependent. (**A**) CD4 cells were activated with anti-CD3 and anti-CD28 mAbs. After 48 hr, cytosolic, nuclear and mitochondrial fractions were prepared and used to examine Stat3 by Western blot analysis. GAPDH and COX IV were used as markers for cytosol and mitochondria, respectively. (**B**) Mitochondrial fractions from freshly isolated CD4 cells, and CD4 cells activated (48 hr) with (IL-6) or without IL-6 (Media) were analyzed for Stat3, phospho-Stat3 (p-Stat3) and the Complex I subunit NDUFA9, as mitochondrial loading control. Relative densitometry ratios of p-Stat3 to NDUFA9 and total Stat3 to NDUFA9 in cells activated in the presence and absence of IL-6 are shown. (**C**) Total Stat3 levels in CD4 cells activated (48 hr) with or without IL-6 were examined by Western blot analysis using whole cell extracts. GAPDH was used as loading control. (**D**) MMP in $Stat3^{+/+}$ or $Stat3^{-/-}$ CD4 cells activated with anti-CD3/28 Abs in the presence or absence of IL-6 for 48 hr (n = 3). (**E**) Percentage of TMRE^high population in $Stat3^{+/+}$ or $Stat3^{-/-}$ CD4 cells activated from (**D**) (n = 3). (**F**) Percentage of TMRE^high population in WT or mut-*Stat3* CD4 cells activated from (n = 3). (**G**) Mitochondrial $Ca^{2+}$ (Rhod-2 AM staining) in $Stat3^{+/+}$ or $Stat3^{-/-}$ CD4 cells activated as in (**D**). (**H**) Percentage of Rhod-2^high population (gate shown in panel G) in $Stat3^{+/+}$ or $Stat3^{-/-}$ CD4 cells (n = 3). (**I**) Mitochondrial $Ca^{2+}$ (Rhod-2 AM staining) in WT or mut-*Stat3* CD4 cells activated as in (**D**). (**J**) Percentage of Rhod-2^high population (gate shown in panel I) in WT or mut-*Stat3* CD4 cells activated as in (**D**) (n = 3). (**K**) Mitochondrial fractions of CD4 cells activated (48 hr) with or without IL-6 were resolved by BN-PAGE. Supercomplexes regions (SC region) of BN-PAGE were excised and analyzed for Stat3, NDUFA9 and NDUFV1 by Western blot analysis. Error bars represent the mean ± SD. *denotes $p < 0.05$, as determined by Student's t test and one-way or two-way ANOVA test. Results are representative of 2–3 experiments.

The following figure supplements are available for figure 5:

**Figure supplement 1**. Stat3 transcription activity is not required for IL-6 to sustain the MMP.

**Figure supplement 2**. Stat3 contributes to elevated cytosolic $Ca^{2+}$ elicited by IL-6.

Only low levels of Stat3 were present in the mitochondrial fraction from freshly isolated CD4 cells (*Figure 5B*). High levels of Stat3 were detected in mitochondria from activated cells, but these levels were further upregulated by IL-6 (*Figure 5B*). In contrast, as a control, the levels of NDUFA9 were not affected by IL-6 (*Figure 5B*). IL-6 did not have an effect on the total levels of Stat3 either, as determined by Western-blot using whole cell lysates (*Figure 5C*). We also examined whether Stat3 in mitochondria was phosphorylated. No phospho-Stat3 was detected in mitochondria from freshly isolated CD4 cells (*Figure 5B*). Phospho-Stat3 was present in mitochondria of activated CD4 cells, but the levels were substantially higher in the presence of IL-6 (*Figure 5B*). Thus, IL-6 promotes the accumulation of Stat3 in mitochondria during CD4 cell activation.

We then investigated whether IL-6 increases MMP in activated CD4 cells through Stat3. CD4 cells from wild-type (*Stat3*$^{+/+}$) mice and T-cell conditional Stat3 knockout (*Stat3*$^{-/-}$) mice (*Takeda et al., 1998*) were activated in the absence or presence of IL-6, and MMP was examined 48 hr later. Interestingly, IL-6 failed to increase MMP in Stat3-deficient CD4 cells during activation (*Figure 5D,E*), indicating that effect of IL-6 on MMP in CD4 cells is dependent on Stat3. To address whether this effect of Stat3 dissociates from its activity as a transcription factor, we used CD4 cells from mice expressing a mutant Stat3 (mut-*Stat3*) carrying a deletion at V$^{463}$ residue (Stat3-$\Delta$463) that prevents DNA binding but does not affect Stat3 phosphorylation (*Steward-Tharp et al., 2014*). This mutation was found in autosomal dominant hyperimmunoglobulin E syndrome (*Holland et al., 2007*; *Minegishi et al., 2007*; *Jiao et al., 2008*). Expression of mut-*Stat3* in mice has been shown to act as dominant-negative and inhibit Stat3 mediated transcription (*Steward-Tharp et al., 2014*). CD4 cells from WT and mut-*Stat3* mice were activated with or without IL-6 and MMP was examined after 48 hr. IL-6 was still able to increase MMP in CD4 cells from mut-*Stat3* mice (*Figure 5F*). In addition, we also tested the effect of Stattic, a well characterized inhibitor of Stat3 that blocks dimerization of Stat3 through phosphor-Tyr$^{705}$ (*Schust et al., 2006*). The presence of Stattic, even at a relatively high concentration (*Schust et al., 2006*), did not affect the MMP in IL-6-treated CD4 cells (*Figure 5—figure supplement 1A and B*). Thus, correlating with the accumulation of Stat3 in mitochondria, the increased MMP in CD4 cells activated in the presence of IL-6 requires Stat3, but it is independent of Stat3-mediated transcription.

Although the presence of Stat3 in mitochondria and its role as regulator of ETC activity has now been widely reported in different cell types, no previous studies have addressed the role of Stat3 in mitochondrial Ca$^{2+}$. To further determine whether IL-6 increases mitochondrial Ca$^{2+}$ through Stat3, we examined mitochondrial Ca$^{2+}$ in *Stat3*$^{+/+}$ and *Stat3*$^{-/-}$ CD4 cells activated in the presence or absence of IL-6. Interestingly, in the absence of Stat3, IL-6 failed to maintain elevated levels of mitochondrial Ca$^{2+}$ (*Figure 5G,H*). To show that this effect was not dependent on Stat3 transcriptional activity we also examined mitochondrial Ca$^{2+}$ in CD4 cells from mut-*Stat3* mice. Unlike Stat3 deficient CD4 cells, IL-6 was capable to increase mitochondrial Ca$^{2+}$ in mut-*Stat3* CD4 cells (*Figure 5I,J*). To further examine whether Stat3 is necessary for the regulation of cytosolic Ca$^{2+}$ elicited by IL-6, cytosolic Ca$^{2+}$ levels were measured in *Stat3*$^{+/+}$ or *Stat3*$^{-/-}$ CD4 cells activated in the presence or absence of IL-6 using the Fura-2 AM assay. Unlike *Stat3*$^{+/+}$ CD4 cells, IL-6 failed to increase cytosolic Ca$^{2+}$ in *Stat3*$^{-/-}$ CD4 cells (*Figure 5—figure supplement 2*). Together, these data show for the first time that Stat3 contributes to mitochondrial Ca$^{2+}$ in response to IL-6 and, consequently, cytosolic Ca$^{2+}$ homeostasis.

Previous studies have demonstrated the association of Stat3 with Complex I of the ETC through GRIM-19, a component of Complex I (*Lufei et al., 2003*; *Gough et al., 2009*; *Wegrzyn et al., 2009*; *Tammineni et al., 2013*). No studies have reported whether Stat3 is present in the ETC SCs. Our studies above (*Figure 2D*) indicate that IL-6 facilitates the formation of ETC SCs in CD4 cells. We therefore examined whether mitochondrial Stat3 could also be recruited to the SCs. BN-PAGE was performed using mitochondrial extracts generated with digitonin from CD4 cells activated in the presence or absence of IL-6. SC region of BN-PAGE was excised and resolved by Western blot analysis for Stat3. As described above, the levels of SCs were increased in CD4 cells activated in the presence of IL-6 as determined by the levels of NDUFA9 and NDUFV1 subunits of Complex I (*Figure 5K*). Interestingly, Stat3 was also present in the SC region isolated from IL-6-treated CD4 cells (*Figure 5K*). Thus, Stat3 is recruited to the ETC SCs, where it can regulate activity of Complex I through interaction with GRIM-19.

## Mitochondrial Ca$^{2+}$ is essential for IL-6 to sustain the production of IL-21 and IL-4 late during activation of CD4 cells

IL-6, in the absence of other cytokines, is the major inducer of IL-21 production by CD4 cells in mouse and human (*Nurieva et al., 2008*; *Suto et al., 2008*; *Dienz et al., 2009*; *Diehl et al., 2012*).

Stat3 is considered the main transcription factor that induces *Il21* gene expression (*Chen et al., 2006*; *Nurieva et al., 2007*; *Zhou et al., 2007*; *Kaplan et al., 2011*). However, since Stat3 but not its transcriptional activity is required for IL-6 to sustain MMP and Ca$^{2+}$ during the activation of CD4 cells, this could be an additional mechanism by which IL-6 promotes the production of IL-21. We therefore examined the ability of IL-6 to induce IL-21 production in CD4 cells from mut-*Stat3* mice where Stat3 is present but its transcriptional activity is impaired. Similarly to human CD4 cells from patients with Hyper IgE syndrome expressing mut-*Stat3*, CD4 cells from mut-*Stat3* mice have been shown to fail to produce IL-17, another cytokine gene regulated by Stat3 (*Ma et al., 2008*; *Milner et al., 2008*; *Renner et al., 2008*; *de Beaucoudrey et al., 2008*; *Minegishi et al., 2009*; *Durant et al., 2010*; *Steward-Tharp et al., 2014*). Although IL-6 totally failed to induce IL-21 production in *Stat3$^{-/-}$* CD4 cells (*Figure 6A*), it was able to trigger the production of IL-21 in mut-*Stat3* CD4 cells (*Figure 6B*). Thus, correlating with its role on MMP and mitochondrial Ca$^{2+}$, Stat3 can contribute to the production of IL-21 in response to IL-6 independently of its function of transcription factor.

A recent study has reported that sustained elevated cytosolic Ca$^{2+}$ levels are associated with the increased expression of *Il21* in CD4 cells in vivo (*Shulman et al., 2014*). We therefore investigated whether the sustained high MMP elicited by IL-6 late during the activation of CD4 cells could contribute to the production of IL-21 triggered by this cytokine. CD4 cells were activated in the presence or absence of IL-6 for 42 hr and treated with rotenone and antimycin (R/A) or CCCP (to depolarize mitochondria) for another 6 hr. IL-21 levels in the supernatants were determined by Enzyme linked immunosorbent assay (ELISA). Although there were already substantial levels of IL-21 at 42 hr in cells activated with IL-6, these levels steeply rose in the next 6 hr (*Figure 6C*). However, the increase in IL-21 levels was prevented by the treatment with R/A or CCCP (*Figure 6C*), indicating that the late production of IL-21 was dependent on the increased MMP caused by IL-6. To further address whether IL-6-mediated mitochondrial Ca$^{2+}$ contributes to the late production of IL-21, CD4 cells were activated in the presence or absence of IL-6 for 42 hr, and treated with CGP-37157 to inhibit mitochondrial Ca$^{2+}$ export for another 6 hr. The increase in IL-21 production was also prevented by CGP-37157 (*Figure 6C*), showing that the increased mitochondrial Ca$^{2+}$ elicited by IL-6 also contributes to the late production of IL-21. Similarly, CGP-37157 prevented the increase in IL-21 production late during activation in mut-*Stat3* CD4 cells, without effecting IL-2 production (*Figure 6—figure supplement 1A and B*).

We and others have shown that IL-6 can also promote the production of IL-4 during activation (*Rincon et al., 1997*; *Diehl et al., 2002*; *Heijink et al., 2002*). Like IL-21, sustained elevated cytosolic Ca$^{2+}$ levels have been associated with the increased expression of *Il4* in CD4 cells in vivo (*Shulman et al., 2014*). We therefore examined the effect that interfering with MMP or Ca$^{2+}$ has on IL-4 production later during activation. Similar to IL-21, the levels of IL-4 were increased in the last 6 hr in IL-6-stimulated CD4 cells, however R/A, CCCP or CGP-37157 prevented this increase (*Figure 6D*), indicating that the increased MMP and cytosolic Ca$^{2+}$ regulated by mitochondrial Ca$^{2+}$ caused by IL-6 also contributes to the late production of IL-4. In contrast, IL-6 had no effect on IL-2 production and treatment with R/A, CCCP or CGP-37157 had no effect (*Figure 6E*). We also examined the relative contribution of transcription-independent function of Stat3 in the regulation of these other cytokines by IL-6. Similar to IL-21, IL-4 production was strongly reduced in *Stat3$^{-/-}$* CD4 cells, but not in mut-*Stat3* CD4 cells (*Figure 6—figure supplement 2*). In contrast, IL-2 production was more affected in mut-*Stat3* CD4 cells than in *Stat3$^{-/-}$* CD4 cells (*Figure 6—figure supplement 2*), further supporting a transcription-independent role of Stat3 in the regulation of IL-21 and IL-4 by IL-6.

To address whether mitochondrial Ca$^{2+}$ could contribute to the IL-6-mediated gene expression of these cytokines, we also examined mRNA levels of *Il21*, *Il4* and *Il2*. CD4 cells were activated in the presence of or absence of IL-6, and treated with CGP-37157 to inhibit mitochondrial Ca$^{2+}$ export. The levels of *Il21* and *Il4* mRNA were significantly increased in cells treated with IL-6 but 6 hr of CGP-37157 treatment was sufficient to reduce these levels (*Figure 6F*). In contrast, *Il2* mRNA levels were not increased by IL-6, and treatment with CGP-37157 did not have an effect. Thus, mitochondrial Ca$^{2+}$ regulated by IL-6 is required for sustaining cytokine gene expression induced by IL-6 in CD4 cells late during activation.

In addition, we also addressed the relevance of mitochondrial Ca$^{2+}$ uptake in the regulation of cytokines by IL-6 using the RU360 compound, a specific MCU inhibitor (*Matlib et al., 1998*).

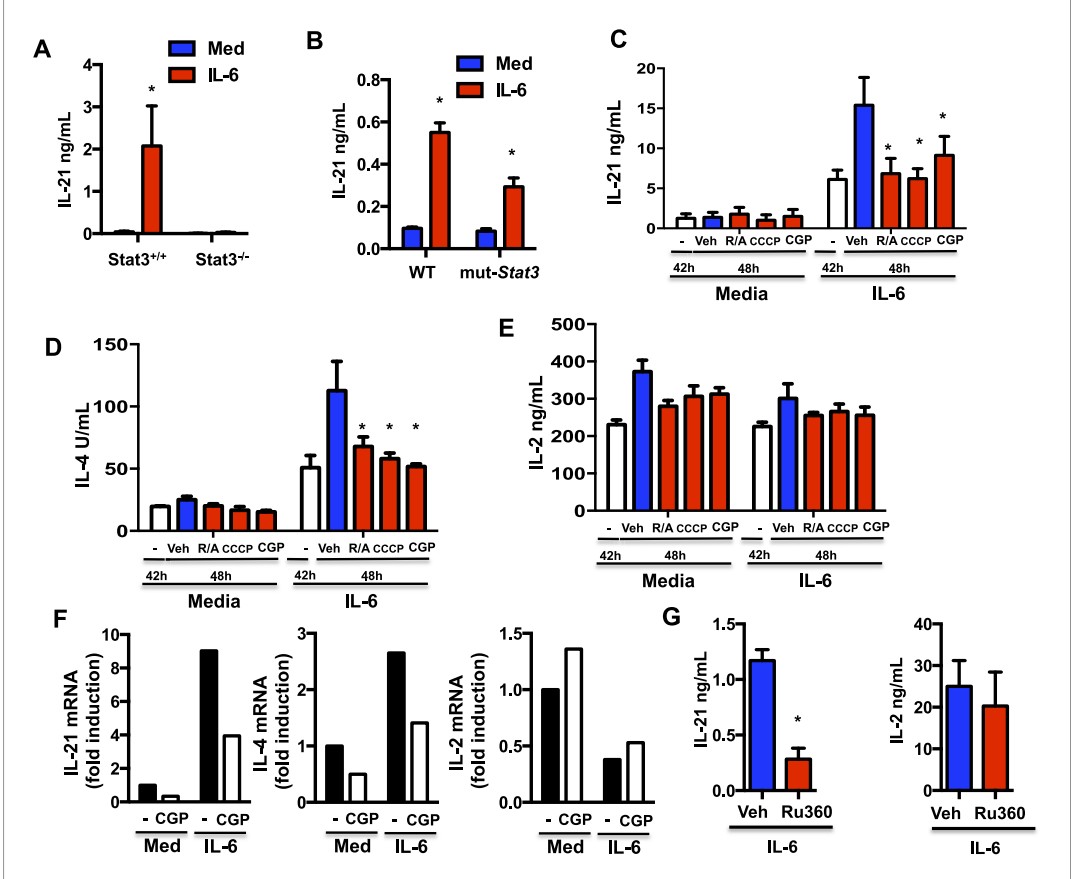

**Figure 6**. Mitochondrial Ca$^{2+}$ is essential for IL-6 to sustain the production of IL-21 and IL-4 late during activation of CD4 cells. (**A**) CD4 cells from *Stat3*$^{+/+}$ or *Stat3*$^{-/-}$ mice were activated in the presence or absence of IL-6 for 48 hr. IL-21 production was measured by ELISA. (**B**) IL-21 production from WT or mut-*Stat3* CD4 cells with or without IL-6 during activation was measured as in (**A**) CD4 cells were activated in the presence or absence of IL-6. After 42 hr, rotenone/antimycin (R/A), CCCP, GCP-37157 or vehicle were added to the cultures. Supernatants were collected 6 hr later. IL-21 (**C**), IL-4 (**D**), IL-2 (**E**) production was measured by ELISA. (**F**) Relative mRNA levels for IL-21, IL-4 and IL-2 in activated in CD4 cells (48 hr) were measured by real-time PCR (RT-PCR). (**G**) CD4 cells were activated in the presence of IL-6. After 24 hr, Ru360 or vehicle control (Veh) were added to the cultures. Supernatants were collected 24 hr later. IL-21 and IL-2 production was measured by ELISA. Error bars represent the mean ± SD. *denotes $p < 0.05$, as determined by two-way ANOVA test. Results are representative of 2–3 experiments.

The following figure supplements are available for figure 6:

**Figure supplement 1**. A transcriptionally inactive Stat3 is sufficient for IL-6 to promote IL-21 production through mitochondrial Ca$^{2+}$.

**Figure supplement 2**. Stat3 contributes to the production of IL-4 in response to IL-6 independently of its function of transcription factor.

**Figure supplement 3**. Ru360 decreases mitochondrial Ca$^{2+}$ in CD4 cells in response to IL-6 during activation.

We confirmed that the treatment with RU360 lowered the mitochondrial Ca$^{2+}$ levels in CD4 cells activated in the presence of IL-6 (*Figure 6—figure supplement 3*). Importantly, the treatment with RU360 reduced the production of IL-21 in CD4 cells activated with IL-6 (*Figure 6G*). RU360 however had no effect on IL-2 production (*Figure 6G*). Thus, both uptake and export of mitochondrial Ca$^{2+}$ plays a role in the regulation of cytokine production by IL-6 in CD4 cells.

# Increased mitochondrial Ca²⁺ elicited by IL-6 is required to sustain nuclear NFAT accumulation late during activation of CD4 cells

*Il21* gene expression is regulated by Stat3, a Ca²⁺-independent transcription factor, but it is also regulated by the NFAT transcription factor (**Kim et al., 2005**; **Durant et al., 2010**). NFAT is also required for *Il4* gene expression (**Rao, 1994**; **Diehl et al., 2002**; **Rengarajan et al., 2002**). Nuclear translocation of NFAT is dependent on increased cytosolic Ca²⁺ and activation of the Ca²⁺-dependent phosphatase, calcineurin. Mitochondrial Ca²⁺ has been shown to contribute to NFAT activation in sensory neurons (**Kim and Usachev, 2009**). Since we have shown IL-6 promotes NFATc2 nuclear accumulation (**Diehl et al., 2002**), we examined whether this could be dependent on mitochondrial Ca²⁺. CD4 cells were activated in the presence of or absence of IL-6 for 42 hr, and treated with CGP-37157 for another 6 hr to inhibit mitochondrial Ca²⁺ export. The addition of CGP disrupted the nuclear accumulation of NFATc2 in cells treated with IL-6 (**Figure 7A**). Thus, mitochondrial Ca²⁺ regulated by IL-6 is required for IL-6 to sustain NFATc2 in the nucleus late during activation.

To address whether NFAT contributes to the production of IL-21 and IL-4 induced by IL-6 late during activation, CD4 cells were activated in the presence or absence of IL-6 for 42 hr, and treated for another 6 hr with FK506, a NFAT inhibitor (FK). FK506 blocked the production of IL-21 and IL-4 induced by IL-6, as determined by ELISA (**Figure 7B**). In addition, inhibition of NFAT late during activation also reduced *Il21* and *Il4* mRNA levels in cells exposed to IL-6 (**Figure 7C**). Therefore, high mitochondrial Ca²⁺ and nuclear accumulation of NFAT triggered by IL-6 late during activation in CD4 cells is required to sustain expression of *Il21* and *Il4*.

## Discussion

Most of the functions of IL-6 in CD4 cells have been assigned to a regulatory role on gene expression through Stat3 as a transcription factor. However, in the light of studies indicating that Stat3 localizes in mitochondria where it regulates the mitochondrial respiratory chain through association with Complex

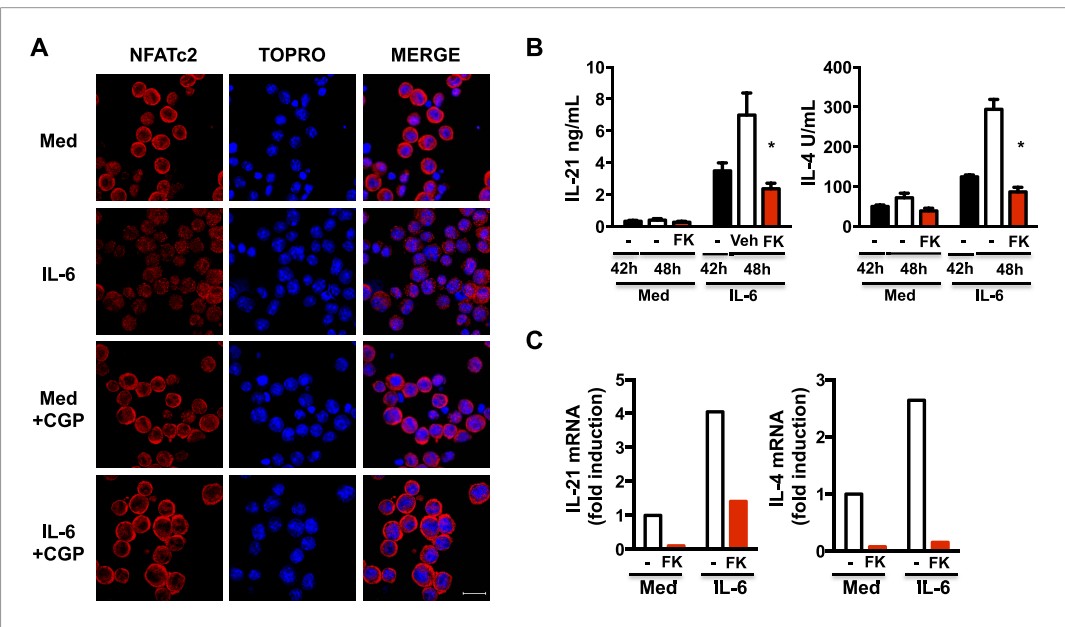

**Figure 7**. Increased mitochondrial Ca²⁺ by IL-6 is required to sustain nuclear NFAT accumulation late during activation of CD4 cells. (**A**) CD4 cells were activated in the presence or absence of IL-6 for 42 hr followed by 6 hr treatment with medium (Med) or CGP-37157 (CGP). NFATc2 (red) was examined by immunostaining and confocal microscopy. TOPRO was used as nuclear dye. 40× Magnification. Bars represent 20 μm. (**B**) CD4 cells were activated in the presence or absence of IL-6 for 42 hr. FK506 (FK) was added to culture for another 6 hr. Supernatants were collected, and IL-21 and IL-4 levels were measured by ELISA. (**C**) Relative mRNA for IL-21 and IL-4 levels in CD4 cells activated as in (**B**) was measured by RT-PCR. Error bars represent the mean ± SD. *denotes p < 0.05, as determined by two-way ANOVA test. Results are representative of 2–3 experiments.

I (*Gough et al., 2009*; *Wegrzyn et al., 2009*), it was also possible that IL-6 could have an effect on mitochondria in CD4 cells. Our studies here show for the first time that IL-6 maintains mitochondrial hyperpolarization late during activation of CD4 cells and this has an impact in mitochondrial $Ca^{2+}$ and, thereby cytosolic $Ca^{2+}$. We also show that the effect of IL-6 on mitochondrial $Ca^{2+}$ and baseline cytosolic $Ca^{2+}$ requires the presence of Stat3, but it is independent of its role as transcription factor.

In recent years, growing interest has been focused on mitochondrial biology in T cells. Bioenergetic profiling of T cells has revealed that T cell metabolism changes dynamically during activation (*Wang and Green, 2012*; *Pearce et al., 2013*). Naive T cells maintain low rates of glycolysis and predominantly oxidize glucose-derived pyruvate via OXPHOS, or engage fatty acid oxidation (FAO). After activation, they rapidly switch to anabolic growth and biomass accumulation. This adaption to aerobic glycolysis is specifically required for effector functions in T cells (*Chang et al., 2013*). IL-2 and IL-15 have been reported to regulate mitochondrial respiration and the balance between glycolysis and oxidative phosphorylation (*van der Windt et al., 2012*). IL-2 has been shown to support aerobic glycolysis, while IL-15 increases spared respiratory capacity and oxidative metabolism by enhancing mitochondrial biogenesis and FAO in CD8 cells (*Pearce et al., 2009*; *van der Windt et al., 2012*). IL-6 has been recently shown to regulate glucose homeostasis in myeloid cells and induce the switch from white adipose tissue to brown fat in cancer induced cachexia (*Mauer et al., 2014*; *Petruzzelli et al., 2014*). Here we show that IL-6 enhances the MMP in CD4 cells. However, this is uncoupled from oxidative phosphorylation (i.e. ATP synthesis). In addition, IL-6 does not alter the balance between glycolysis and oxidative glycolysis during activation. Instead, we show that a sustained MMP elicited by IL-6 leads to an effect on mitochondrial $Ca^{2+}$. No other studies have linked cytokine effects to mitochondrial $Ca^{2+}$ in CD4 cells.

While endoplasmic reticulum (ER)-derived $Ca^{2+}$ has been extensively studied in T cells, less is known about mitochondrial $Ca^{2+}$ homeostasis in T cells. Mitochondrial $Ca^{2+}$ has been previously shown to modulate store-operated calcium signaling early upon T cell activation at the immunological synapse (*Hoth et al., 1997*; *Quintana et al., 2007*). Here we show that IL-6 uses MMP to sustain elevated levels of mitochondrial $Ca^{2+}$ late during activation and, consequently, elevated levels of cytosolic $Ca^{2+}$. We have previously shown that the expression of IP$_3$Rs is downregulated during the activation of CD4 cells (*Nagaleekar et al., 2008*). It is therefore possible that the source of $Ca^{2+}$ in CD4 cells is reprogramed during activation. ER- IP$_3$R is the main source of $Ca^{2+}$ during early activation of naive CD4 cells at the synapse. However, mitochondrial $Ca^{2+}$ could be the major source to sustain cytosolic $Ca^{2+}$ in activated CD4 cells. Our data indicate that IL-6 sustains cytosolic $Ca^{2+}$ late during activation by increasing the MMP and mitochondrial $Ca^{2+}$. This provides a potential mechanism by which Tfh cells have increased free cytosolic calcium levels (*Shulman et al., 2014*). More importantly, we show here for the first time that mitochondrial $Ca^{2+}$ plays a key role in promoting increased production of cytokine by effector CD4 cells. Although IP$_3$R-mediated $Ca^{2+}$ release is essential for the initial induction of cytokine gene expression (*Feske, 2007*), we have previously shown that IP$_3$R-mediated $Ca^{2+}$ is not responsible for late production of cytokines by activated CD4 cells (*Nagaleekar et al., 2008*). Thus, the source of $Ca^{2+}$ for cytokine production is also reprogrammed during activation of CD4 cells. Although we cannot discard the effect of other transcription factors, our study shows that mitochondrial $Ca^{2+}$ is required for IL-6 to keep NFATc2 in the nucleus, and that NFAT contributes to late expression of *Il21* and *Il4*.

Mitochondrial respiration has been shown to lead to ROS production caused by proton leaks and ROS can lead to oxidative injury. A number of recent studies have shown that mROS can function as signaling intermediates, and the mROS signaling is required for antigen-specific T cell activation and subsequent IL-2 production (*Byun et al., 2008*; *Schieke et al., 2008*; *Sena et al., 2013*). Although IL-6 increases the MMP, we did not observe an increase in the levels of mROS correlating with the effect of IL-6 facilitating the formation of respiratory SCs. The presence of ETC SCs is emerging as a novel but highly relevant aspect of the mitochondrial function (*Acín-Pérez et al., 2008*; *Althoff et al., 2011*). The function of these SCs is to facilitate the transfer of electrons between ETC complexes to minimize the risk of electron leak and, thereby, the risk of producing harmful ROS. Our study demonstrates for the first time the presence of ETC SCs in CD4 cells, and the effect that IL-6 has in promoting the formation of these SCs during activation of CD4 cells. This could be a mechanism by which IL-6 can sustain elevated MMP and $Ca^{2+}$ while minimizing the production of mROS. Although the association of Stat3 with individual complexes of the ETC has been previous described in heart and cancer cells (*Gough et al., 2009*; *Wegrzyn et al., 2009*), here we show for the first time the

presence of Stat3 in the ETC SCs in CD4 cells. Stat3 may also be present in mitochondrial SCs in other tissues such as heart.

Thus, here we identify a novel mechanism by which IL-6 promotes the production of IL-21 and IL-4 late during the activation of CD4 cells. This new mechanism involves Stat3 but as a factor regulating MMP and $Ca^{2+}$ instead of its function as mediator of transcription. Our studies also reveal a novel function of mitochondrial respiration in the control of cytokine production through its effect on mitochondrial $Ca^{2+}$ homeostasis.

## Materials and Methods

### Mice

C57BL/6J mice were purchased from Jackson Laboratories. Null IL-6 deficient mice (IL-6 KO) were previously described (*Poli et al., 1994*). Stat3 conditional knockout (*Stat3*$^{-/-}$) mice were generated by crossing the homozygous floxed Stat3 mice (*Stat3*$^{loxp/loxp}$) (*Takeda et al., 1998*) with T cell-specific *Lck*-Cre transgenic [B6.Cg-Tg(*Lck*-cre)1Cwi N9] mice (*Lee et al., 2001*). Mutant-Stat3 (mut-Stat3) mice have been previously described (*Steward-Tharp et al., 2014*). OT-II TCR transgenic mice have been previously described (*Barnden et al., 1998*). All mice were housed under sterile conditions at the animal care facility at the University of Vermont. All procedures performed on the mice were approved by the University of Vermont Institutional Animal Care and Use Committee.

### Cell purification and activation *in vitro*

CD4 cells were isolated from spleen and lymph nodes by negative selection as previously described (*Diehl et al., 2002*). For *Stat3*$^{+/+}$ and *Stat3*$^{-/-}$ mice, CD4 cells were purified by cell sorting (FACS-Aria; Becton Dickinson). CD4 cells were activated with plate-bound anti-CD3 (2C11) (5 $\mu$g/ml) and soluble anti-CD28 (1 $\mu$g/ml) (BD Pharmingen, San Diego, CA) mAbs in the presence or absence of IL-6 (50 ng/mL) (Miltenyi Biotec, Auburn, CA). Pharmacological inhibitors were added to culture 42 hr after activation and supernatants were harvested 6 hr later. APCs were purified by depleting CD4 and CD8 T cells using positive selection (Miltenyi), and followed by irradiation treatment (2000 rad). APCs and OT-II CD4 cells were co-cultured at 4:1 ratio in the presence of 5 $\mu$M OVA$_{323-339}$ peptide (*Barnden et al., 1998*) with or without IL-6 (50 ng /mL) (Miltenyi) or anti-IL-6 (2.5 $\mu$g /mL) (BD Pharmingen).

Pharmacological inhibitors used were CGP-37157 (Tocris Bioscience, Ellisville, MO) (10 $\mu$M), CCCP (2 $\mu$M), rotenone (2 $\mu$M), antimycin (2 $\mu$M), Ru360 (10 $\mu$M), FK506 (InvivoGen, San Diego, CA) (10 nM), Stattic (10 $\mu$M).

### Immunization experiment in vivo

OT-II CD4 cells were purified from OT-II TCR transgenic mice (Thy1.1$^+$) by positive selection using anti-CD4 MACS beads (Miltenyi Biotec). $2 \times 10^6$ naive OT-II TCR Tg T cells in 100 $\mu$L Phosphate buffered saline (PBS) were transferred i.v. into WT or IL-6 KO hosts (Thy1.2$^+$). After overnight, adoptive hosts were simultaneously immunized i.p. with 200 $\mu$L of 50 $\mu$g OVA absorbed on alum (4.5%, w/v). After 2 d immunization, spleens from immunized mice were harvested and stained with fluorescent conjugated Abs (anti-Thy1.1, anti-V$\alpha$2, anti-CD69, anti-CD4, anti-CD44) and TMRE followed by flow cytometry analysis. For each experiment, three to four hosts were used in each group.

### Flow cytometry analysis

MMP analysis was performed by staining CD4 cells with TMRE (Molecular Probes, Eugene, OR) as previously described (*Hatle et al., 2013*). Mitochondrial calcium analysis was performed by staining with Rhod-2 AM (Invitrogen, Carlsbad, CA; 5 or 10 $\mu$M) for 1 hr at 37°C, as previously described (*Brisac et al., 2010*). mROS production was determined by 10 min staining of cells with 5 $\mu$M MitoSox Red (Molecular Probes). Live/dead cell viability staining (Molecular Probes) was performed as recommended by the manufacturer. All samples were examined by flow cytometry analysis using an LSRII flow cytometer (BD Biosciences) and Flowjo software.

### Western blot analysis

Whole-cell extracts were prepared in Triton lysis buffer. Mitochondrial, nuclear and cytosolic extracts were purified using the cell fractionation kit-standard (MitoScience) for CD4 cells. Western blot analyses were performed as previously described (*Hatle et al., 2013*). Anti-Stat3, anti-phospho-Stat3 (Tyr705)

(Cell Signaling, Danvers, MA), anti-actin, anti-GAPDH, anti-rabbit IgG, and anti-goat IgG (Santa Cruz Biotechnology, Santa Cruz, CA); anti-mouse IgG (Jackson Immunologicals, West Grove, PA); anti-CoxIV (Cell Signaling); anti-NDUFA9, anti-NDUFS3 (MitoScience, Eugene, OR) Abs were used.

## Electron microscopy imaging

Cells were suspended in fixative for 60 min at 4°C (2% glutaraldehyde, 0.05% CaCl$_2$, 0.1% MgCl$_2$, 22 mM betaine in 0.1 M Pipes buffer). After rinsing in Pipes buffer, the cell pellets were embedded in 2% SeaPrep agarose, crosslinked with above fixative and postfixed with 1% osmium tetroxide for 1 hr at 4°C. The samples were again rinsed in Pipes buffer, followed by dehydration through graded ethanol, cleared in propylene oxide and embedded in Spurr's epoxy resin. Semithin sections (1 $\mu$m) were cut with glass knives on a Reichert ultracut microtome, stained with methylene blue-azure II, and evaluated for areas of interest. Ultrathin sections (60–80 nm) were cut with a diamond knife, retrieved onto 200 mesh thin bar nickel grids, contrasted with uranyl acetate (2% in 50% ethanol) and lead citrate, and examined with a JEOL 1400 TEM (JOEL USA Inc, Peabody, MA) operating at 60 kV. Twenty-five digital images were acquired with an AMT XR611 CCD camera by systemic uniform random sampling from each sample. Number of mitochondria and mitochondria with tight or expanded cristae was counted manually.

## Confocal microscopy analysis

Activated CD4 cells (48 hr) were cytospun and immunostained as previously described (*Diehl et al., 2002*) using a specific anti-NFATc2 Ab (Upstate Biotechnology, Lake Placid, NY), followed by Alexa568-conjugated secondary Ab. Nuclei were stained with TOPRO (Molecular Probes). Images were recorded using a Zeiss LSM 510 Meta confocal laser scanning imaging system (Carl Zeiss Microimaging, Thornwood, NY).

## Blue-native PAGE

Purified mitochondria were solubilized in Native PAGE loading buffer (Invitrogen) containing 2% digitonin (Sigma-Aldrich Co., St Louis, MO). Complexes were resolved by native electrophoresis through gradient 4–16% Native PAGE Novex Bis-Tris gels (Invitrogen) as previously described (*Hatle et al., 2013*). Proteins were transferred to PVDF membrane for Western blot analysis with anti-NDUFA9 (MitoScience) and anti-Core I (MitoScience). SCs regions were also excised from BN-PAGE, eluted in SDS sample buffer and resolved in SDS-PAGE. Proteins were then transferred to PVDF membrane for Western blot analysis with anti-NDUFA9, anti-NDUFV1 and anti-Stat3 Abs.

## Mitochondrial respiration and extracellular acidification

OCR were measured, as previously described (*van der Windt et al., 2012*) under basal conditions and in response to oligomycinv (1 $\mu$M), FCCP (1 $\mu$M), and rotenone + antimycin A (1 $\mu$M) with the Seahorse XF-24 Extracellular Flux Analyzer (Seahorse Bioscience, North Billerica, MA) using the XF Cell Mito Stress Test Kit. ECAR were measured as recommended by the manufacturer using the XF Glycolysis Stress Test Kit.

## RNA isolation and RT-PCR

Total RNA was isolated from CD4 cells using the Qiagen micro RNeasy kit, as recommended by manufacture (Qiagen, Valencia, CA). cDNA synthesis was performed as previously described (*Hatle et al., 2013*). cDNA was used to quantify the relative mRNA levels for mouse *Il21*, *Il4* and *Il2* (Assays-on-Demand by Applied Biosystems) by conventional RT-PCR (Applied Biosystems, San Diego, CA) using β2-microglobulin as housekeeping gene. The relative values were determined by the comparative CT analysis method.

## ELISA

Cytokine levels in cell culture supernatants were determined by ELISA as previously described (*Diehl et al., 2002*; *Dienz et al., 2007*, *2009*).

## ATP and Lactic acid measurement

ATP was measured on 10$^5$ cells and/or 100 µl of culture supernatants by using ATP Lite kit (Perkin Elmer, Boston, MA) as recommended by the manufacturer in a TD-20/20 single tube luminometer. Lactate production was examined in CD4 cells (2 × 10$^6$) activated for 48 hr, washed and incubated for 2 hr in media. Measurement of lactate in supernatants was done using the Lactate assay Kit II (BioVision, Milpitas, CA).

## Cytosolic calcium measurement

Cytosolic calcium was measured by staining with Fura-2 AM (Molecular Probes) (5 $\mu$M) for 30 min, followed by fluorometrically measurement (340/380 exication, 510 emission) in a Synergy H4 plate reader (Bio-Tek, Winooski, VT). $F_{340/380}$ value was calculated by dividing the fluorescence reading at 340 nm by the fluorescence at 380 nm exication. Cells were also loaded for 45 min at 37°C with 10 $\mu$M Indo-1 (Molecular probes) (*Grynkiewicz et al., 1985*), harvested, washed and transferred to a standard extracellular solution (140 mM NaCl, 4 mM KCl, 1 mM $CaCl_2$, 2 mM $MgCl_2$, 1 mM $KH_2PO_4$, 10 mM glucose, 10 mM HEPES [pH 7.4]). The ratio of $Ca^{2+}$-bound Indo-1 fluorescence (405 nm) to unbound indo-1 fluorescence (480 nm) was then determined by flow cytometry analysis.

## Statistical analysis

Significance of differences between two groups was determined using GraphPad Prism v. 5.0, by standard Student's t-test. Significance of differences among more than 2 groups was determined by one-way or two-way ANOVA. Standard $p < 0.05$ was used as the cutoff for significance. For flow cytometry analysis, percentages of compared samples under the same gate were analyzed by t-test or ANOVA in Prism.

## Acknowledgements

We thank N. Bishop for expertise and helpful discussions on the confocal images and TEM images, T. Hunter, J. Hoffman and M. Shane for help with real-time PCR analysis, and R. del Rio-Guerra for help with flow cytometry analysis (University of Vermont, Burlington, VT). We would like to thank Dr. John O'Shea (NIH/NIAMS) for kindly providing the mut-*Stat3* mice, Phani Gummadidala for performance of some of the experiments, and Brian Silverstrim for technical support (University of Vermont, Burlington, VT). This work was supported by National Institutes of Health grant R56AI094027 (to M.R.), P20 GM103496 (to M.R. and R.Y.) and RR019246-01 (to D.J.T.).

## Additional information

### Funding

| Funder | Grant reference | Author |
| --- | --- | --- |
| National Institutes of Health (NIH) | R56AI094027 | Mercedes Rincón |
| National Institutes of Health (NIH) | P20GM103496 | Rui Yang, Mercedes Rincón |
| National Institutes of Health (NIH) | RR019246-01 | Douglas J Taatjes |

The funders had no role in study design, data collection and interpretation, or the decision to submit the work for publication.

### Author contributions

RY, DL, MR, Conception and design, Acquisition of data, Analysis and interpretation of data, Drafting or revising the article; TMT, DMJ-G, SAD, LKC, DJT, Acquisition of data, Analysis and interpretation of data, Drafting or revising the article; MM, Analysis and interpretation of data, Drafting or revising the article, Contributed unpublished essential data or reagents; CT, LH, Analysis and interpretation of data, Drafting or revising the article

### Ethics

Animal experimentation: All procedures performed on the mice were approved by the Institutional Animal Care and Use Committee (IACUC) of University of Vermont using protocols #12-032 (Rincon), #11-024 (Teuscher) and by the IACUC of Trudeau Institute using protocol #03-005 (Haynes).

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
