## [Decision Letter]

Thank you for sending your work entitled “Mitochondrial Ca^2+^ and membrane potential, an alternative pathway for IL-6 to regulate CD4 cell effector function” for consideration at *eLife*. Your article has been favorably evaluated by Tadatsugu Taniguchi (Senior editor), a Reviewing editor, and two reviewers.

The Reviewing editor and the reviewers discussed their comments before we reached this decision, and the Reviewing editor has assembled the following comments to help you prepare a revised submission.

Calcium signaling is a critical factor involved in T cell activation, differentiation and function. Therefore, understanding of the regulatory mechanisms involved in governing these procedures is an interesting and important topic for controlling T cell responses. In this study, Rui Yang et al. present an interesting finding that IL6 affects CD4 T cell effector functions, especially IL-21 and IL-4 production, via the regulation of mitochondrial calcium and membrane potential, a molecular link that has not been explored before. Specifically, the data here show that IL-6 promotes mitochondrial supercomplex assembly and high mitochondrial membrane potential (MMP) in a mechanism that is uncoupled from mitochondrial ATP production by oxidative phosphorylation. Rather, IL6 maintains MMP during activation of CD4 cells by increasing the number of mitochondria with tight and organized cristae. Although this study provides some interesting findings, the mechanism(s) could be more clearly delineated and better support for the relevance of mitochondrial calcium storage in T cells would strengthen this study. Also, further addressing the mechanism of modulation of mitochondrial function by which Stat3 regulates IL-4 and IL-21 gene expression downstream of IL-6 would be of outstanding interest and extend the scope of the study. Below are several comments for the authors to consider:

1) Since mitochondrial activity is critical to sustain MCU activity to facilitate calcium uptake into mitochondria, inhibiting mitochondrial activity by CCCP reduced mitochondrial calcium levels. However, cytosolic calcium level of CCCP-treated cells also dropped. On the other hand, treatment with mitochondrial calcium exporter inhibitor CGP-37157 suppressed cytosolic calcium level. These findings are difficult to interpret: whether the cytosolic calcium level is controlled by mitochondrial calcium uptake or release or through the coordination of both import and release. This point is not clear and the experimental support is lacking. The authors also should monitor the mitochondrial calcium level in CGP-37157 treated cells to confirm the changes in mitochondrial calcium level.

2) Similar to point #1, the authors should determine the total calcium level in IL-6 treated T cells in the presence or absence of CCCP. It is possible that mitochondrial calcium uptake has to be increased in IL-6 treated cell to prevent the net loss of intracellular calcium, which can be detrimental for initiating IL-21 and IL-4 expression. Moreover, the authors should examine the effect of RU360 (MCU inhibitor) on IL6-treated CD4 T cells. This approach will be more specific to examine how mitochondrial calcium uptake affects the effector functions of IL6-treated cells because mitochondrial membrane potential affected by CCCP and antimycin cannot address this point.

3) After IL-6 treatment, CD4 T cells increased mitochondrial membrane potential and the formation of respiratory chain complexes. However, there is no difference on ECAR and OCR activities. This finding is interesting, but it is not clear whether IL6 treatment leads to changes on mitochondrial number and size, which can affect the readout of mitochondrial membrane potential and ROS production. The authors need to stain the cells with MitoGreen and MitoDeepRed to examine these parameters. Moreover, it may be interesting to analyze the protein expression of mitochondrial complexes and metabolite profile in control and IL6 treated CD4 T cells. These approaches may allow them to determine the underlying mechanisms that affect the MMP but not ECAR and OCR levels.

4) For the cytosolic calcium measurement, the authors measured the FURA-2 intensity. However, the kinetics of calcium accumulation upon TCR stimulation should be measured (including both baseline and stimulated conditions). IL-6 treatment may affect both baseline (as showed in this manuscript) and the activation kinetics due to the regulation provided by mitochondrial calcium uptake and release.

5) In Figure 5, mutant-STAT3 has been used to test whether the DNA-binding ability of STAT3 is required for affecting mitochondrial membrane potential. However, the information of this mutant is lacking. Moreover, the *Tyr705* mutant should be included to examine whether the IL-6 signaling induced STAT3 phosphorylation and dimerization can be involved in affect CD4 T cell mitochondrial activity.

6) Figure 5, the authors stated that IL-6 promotes the accumulation of Stat3 in mitochondria during CD4 T cell activation. However, is the effect on mitochondrial cristae organization and super-complex assembly dependent of the presence of Stat3 in mitochondria? The authors should perform experiments with Stat3 KO and *Stat3*-mut to determine whether the effect of IL-6 on mitochondrial cristae shape and mitochondrial supercomplex assembly is Stat3-dependent and/or depends on their relocation to mitochondria.

7) In the same line, does R/A, CCCP or CGP treatment (Figure 6) in IL-6 activated T cells affect Stat3 positioning to mitochondria?

8) To unequivocally demonstrate the role of Stat3 in the regulation of IL-21 and IL-4 gene expression, the authors should increase cytosolic calcium levels in Stat3 KO and *Stat3* mut T cells and assess IL-21 and IL-4 gene expression upon IL-6 activation.

Also, it would be important if the authors could show that the production of IL-21 and IL-4 in vivo was able to occur in the mutSTAT3 or mito-STAT3 T cells. I think this study highlights a novel aspect of IL-6 signaling and pushes the mechanism to a good extent in vitro, but they have little reconfirmation of their findings in vivo.

Combining the data from the O'Shea paper where the mut-*Stat3* mouse is first described, with the current study, the findings suggest that mito-STAT3 is needed for IL-21 and IL-4 (as described in this paper), but nuclear STAT3 is required for IL-17. It would be great if the authors could better validate this split in STAT3 functions for these cytokines downstream of IL-6. I think this would provide an important distinction in how these cytokines are regulated in the T cells, which is implied by the studies here, but not formally demonstrated.

---

## [Author Response]

*1) Since mitochondrial activity is critical to sustain MCU activity to facilitate calcium uptake into mitochondria, inhibiting mitochondrial activity by CCCP reduced mitochondrial calcium levels. However, cytosolic calcium level of CCCP-treated cells also dropped. On the other hand, treatment with mitochondrial calcium exporter inhibitor CGP-37157 suppressed cytosolic calcium level. These findings are difficult to interpret: whether the cytosolic calcium level is controlled by mitochondrial calcium uptake or release or through the coordination of both import and release. This point is not clear and the experimental support is lacking. The authors also should monitor the mitochondrial calcium level in CGP-37157 treated cells to confirm the changes in mitochondrial calcium level*.

We apologize for the lack of clarity in describing the overall model regarding the contribution of mitochondrial calcium in the homeostasis of cytosolic calcium. We now provide a better description (in the subsection headed “IL-6-mediated high MMP results in elevated mitochondrial Ca^2+^ levels”) summarizing what is known and how our data are integrated in this model. Briefly, previous studies have shown that mitochondria uptake calcium through microdomains formed between mitochondria and plasma membrane or between mitochondria and ER. This uptake of calcium is promoted by maintaining high mitochondrial membrane potential. Then, mitochondria release calcium to the cytosol and regulate more local cytosolic calcium homeostasis. Thus, both uptake and release of mitochondrial calcium are key. For instance, early during activation of T cells, it has been shown that mitochondria relocate to the synapsis and contribute to the uptake of extracellular calcium. We now provide a better description of this model.

In addition, following the suggestion from the reviewers, we now provide new data showing that inhibition of calcium release by CGP-37157 results in increased mitochondrial calcium as expected (new Figure 4—figure supplement 1).

*2) Similar to point #1, the authors should determine the total calcium level in IL-6 treated T cells in the presence or absence of CCCP. It is possible that mitochondrial calcium uptake has to be increased in IL-6 treated cell to prevent the net loss of intracellular calcium, which can be detrimental for initiating IL-21 and IL-4 expression. Moreover, the authors should examine the effect of RU360 (MCU inhibitor) on IL6-treated CD4 T cells. This approach will be more specific to examine how mitochondrial calcium uptake affects the effector functions of IL6-treated cells because mitochondrial membrane potential affected by CCCP and antimycin cannot address this point*.

Since CCCP depolarizes mitochondria we expected a similar effect to the effect obtained with the combination of Rotenone (inhibitor of Complex I) and Antimycin (inhibitor of Complex III) that we showed in Figure 4. Following the recommendations from the reviewers, we now show (new Figure 4—figure supplement 1) that treatment with CCCP suppresses the increased cytosolic calcium levels in IL-6 treated cells, further supporting the contribution of mitochondria to the increased cytosolic calcium in these cells.

Addressing the experiments suggested by the reviewers using RU360 (MCU inhibitor) has been a major challenge since this inhibitor cannot be found commercially anymore. We tried several distributors but it was indefinitely back ordered. Our only alternative was to establish a collaboration with other experts in the field who have this inhibitor. We have now established a collaboration with Dr. Muniswamy Madesh (an expert in mitochondrial calcium uniporter, MCU) we obtained the RU360 inhibitor and performed the suggested experiments. We show now that RU360 treatment reduces mitochondrial calcium in IL-6-treated CD4 cells (new Figure 6—figure supplement 3). In addition, we have also tested the effect of RU360 in cytokine production and we now show that it inhibits the production of IL-21 in IL-6-treated CD4 cells (new Figure 6). We thank the reviewers for the suggestions, although it was challenging to obtain this inhibitor, these data clearly enhance the quality of our study. Dr. Madesh is now a new co-author of our study.

*3) After IL-6 treatment, CD4 T cells increased mitochondrial membrane potential and the formation of respiratory chain complexes. However, there is no difference on ECAR and OCR activities. This finding is interesting, but it is not clear whether IL6 treatment leads to changes on mitochondrial number and size, which can affect the readout of mitochondrial membrane potential and ROS production. The authors need to stain the cells with MitoGreen and MitoDeepRed to examine these parameters. Moreover, it may be interesting to analyze the protein expression of mitochondrial complexes and metabolite profile in control and IL6 treated CD4 T cells. These approaches may allow them to determine the underlying mechanisms that affect the MMP but not ECAR and OCR levels*.

We agree with the reviewers that is an important point. We already addressed it in our first submission with some of the experiments proposed by the reviewers, but we apologize since we did not make major emphasis and it was probably missed. In Figure 1 of the initial version we showed that the levels of NDUFA9 and NDUFS3 (two independent subunits of Complex I) as well as Cox IV (a subunit of Complex IV) are similar between CD4 cells activated in the presence or absence of IL-6. Since there was only one brief sentence with no conclusions, we have now rewritten the section (“IL-6 is essential to sustain mitochondrial membrane potential during activation of 107 CD4 cells”) to conclude that there was no difference in mitochondria content.

Nevertheless, using TEM studies, we now show that the number of mitochondria in IL-6 treated cells is not increased relative to cells activated without IL-6 (new Figure 2—figure supplement 1).

*4) For the cytosolic calcium measurement, the authors measured the FURA-2 intensity. However, the kinetics of calcium accumulation upon TCR stimulation should be measured (including both baseline and stimulated conditions). IL-6 treatment may affect both baseline (as showed in this manuscript) and the activation kinetics due to the regulation provided by mitochondrial calcium uptake and release*.

The kinetics of cytosolic calcium release upon TCR stimulation in CD4 cells activated in the absence or presence of IL-6 is an interesting point and obvious question to ask, primarily if not aware of one of our previous published studies (62). A number of years ago, addressing whether calcium flux was increased in CD4 cells activated with IL-6 because we had previously shown that IL-6 increased NFAT transcriptional activity we obtained novel and unexpected results. Using flow cytometry analysis as the standard method to examine calcium flux in T cells, we found that stimulation with anti-CD3 Ab failed to induce calcium flux in activated CD4 cells independently whether IL-6 was present or not. We went further and showed that this is in part due to the downregulation of IP_3_R expression during activation, and as a result no calcium flux is observed, in contrast to the calcium flux detected in naïve CD4 cells (62). However, although no calcium flux could be triggered by anti-CD3 treatment, we observed that the cytosolic calcium baseline detected by flow cytometry analysis was higher in CD4 cells activated with IL-6, as the reviewers predicted. We did not follow these studies further, but they led us to an evidence to identify the potential role for the sustained mitochondrial membrane potential in CD4 cells activated in the presence of IL-6. We have now included data showing the increased cytosolic calcium baseline in IL-6-activated CD4 cells (analyzed by flow cytometry analysis), while the levels reached by ionomycin (maximum levels) or EGTA (calcium blocker) are comparable (new Figure 4—figure supplement 1).

*5) In*
Figure 5*, mutant-STAT3 has been used to test whether the DNA-binding ability of STAT3 is required for affecting mitochondrial membrane potential. However, the information of this mutant is lacking. Moreover, the* Tyr705 *mutant should be included to examine whether the IL-6 signaling induced STAT3 phosphorylation and dimerization can be involved in affect CD4 T cell mitochondrial activity*.

We apologize for the insufficient information about the DNA binding mutant. In the revised version we now provide a more detailed description of this mutant prior to the description of the experiments we performed (in the subsection headed “The regulation of the MMP and mitochondrial Ca^2+^ elicited by IL-6 is Stat3 dependent”).

To address the contribution of the STAT3 phosphorylation/dimerization in mitochondrial activity we now provide data showing that Stattic, a pharmacological inhibitor of STAT3 that prevents dimerization of STAT3 through P-Tyr^705^, does not prevent the increase in mitochondrial membrane potential caused by IL-6 (new Figure 5—figure supplement 1). In addition, we have tried a large range of concentrations for this inhibitor, but no effect in MMP was found. Thus, STAT3 is required for IL-6 to sustain mitochondrial membrane potential, but STAT3 dimerization is not a required.

*6)*
Figure 5*, the authors stated that IL-6 promotes the accumulation of Stat3 in mitochondria during CD4 T cell activation. However, is the effect on mitochondrial cristae organization and super-complex assembly dependent of the presence of Stat3 in mitochondria? The authors should perform experiments with Stat3 KO and* Stat3*-mut to determine whether the effect of IL-6 on mitochondrial cristae shape and mitochondrial supercomplex assembly is Stat3-dependent and/or depends on their relocation to mitochondria*.

The association of Stat3 with specific complexes of the ETC has been shown by the first two studies describing Stat3 in mitochondria (30; 97). However, the reviewers are correct that there is no evidence for Stat3 being associated with supercomplexes. To address this question we embarked on challenging studies to test if Stat3 is present in supercomplexes in CD4 cells. We have now performed experiments using digitonin-generated mitochondrial extracts from CD4 cells activated in the presence or absence of IL-6, and BN-PAGE to resolve the supercomplexes. The supercomplex region was then excised and protein eluted. The eluted protein was resolved by standard SDS PAGE, and western blot analysis was used to detect Stat3. The supercomplex region was then, eluted and used to run another electrophoresis gel followed by Western blot analysis to test for Stat3. The results show clearly the presence of Stat3 in supercomplexes primarily in those isolated from CD4 cells activated in the presence of IL-6 (new Figure 5). In correlation with the data we had in the first version these data also show an accumulation of supercomplexes in IL-6-activated CD4 cells as determined by the levels of Complex I subunits and Complex IV subunit. These results will represent the first demonstration that Stat3 is indeed recruited to the ETC supercomplexes.

Based on previous published studies, mitochondrial cristae shape can influence the formation of the mitochondrial supercomplexes, but we are not aware of evidence to support that the formation of the supercomplexes could influence the shape of cristae. We therefore think that most likely the presence or absence of Stat3 should not affect the cristae shape. Nevertheless, we have performed TEM for *Stat3*^*-/-*^ and mut-*Stat3* CD4 cells activated in the presence of IL-6. Relative to WT CD4 cells, there were no statistical differences in the distribution of cristae shape in either *Stat3*^*-/-*^ or mut-*Stat3* CD4 cells. We provide here the results for the perusal of the reviewers (Figure 8). We thought that including these data in the manuscript could disrupt the flow of the manuscript. However, if the reviewers and/or editors believe these data are important for the study, we will be happy to incorporate it to the main manuscript.

Author response image 1.**DOI:**
http://dx.doi.org/10.7554/eLife.06376.019

*7) In the same line, does R/A, CCCP or CGP treatment (*Figure 6*) in IL-6 activated T cells affect Stat3 positioning to mitochondria?*

Stat3 is a known upstream regulator of the ETC and mitochondrial membrane potential (MMP) and this is what we have addressed in our studies (role of Stat3 on MMP in IL-6-activated CD4 cells). However, to our knowledge there are no evidences to support a role of MMP in regulating Stat3 localization in mitochondria. While this is an interesting alternative scenario, we believe that this would better be the subject for a follow up study. If the reviewers still feel these experiments are critical to support any of our current conclusions, we can perform it.

*8) To unequivocally demonstrate the role of Stat3 in the regulation of IL-21 and IL-4 gene expression, the authors should increase cytosolic calcium levels in Stat3 KO and* Stat3 *mut T cells and assess IL-21 and IL-4 gene expression upon IL-6 activation*.

We have performed those experiments using ionomycin, a standard calcium ionophore, but we could not rescue the deficiency in cytokines as we predicted. We believe that this is due to the fact that the increase in cytosolic calcium mediated by mitochondria (more local and potentially restricted to subcellular areas) may not be mimicked by the cytosolic calcium increase obtained with ionophores.

However, because of the reviewers’ point, we realize that we did not have evidence to support a role for Stat3 in cytosolic calcium. Therefore, we have now performed experiment to address whether the increase in cytosolic calcium caused by IL-6 is mediated by Stat3. We show now that Stat3 is required for IL-6 to sustain elevated levels of cytosolic calcium (new Figure 5—figure supplement 2). In addition, we also show now that the late production of IL-21 in CD4 cells from mut-*Stat3* mice activated in the presence of IL-6 is dependent on mitochondrial calcium as it happens in WT CD4 cells (new Figure 6—figure supplement 1).

*Also, it would be important if the authors could show that the production of IL-21 and IL-4 in vivo was able to occur in the mutSTAT3 or mito-STAT3 T cells. I think this study highlights a novel aspect of IL-6 signaling and pushes the mechanism to a good extent in vitro, but they have little reconfirmation of their findings in vivo*.

*Combining the data from the O'Shea paper where the mut-*Stat3 *mouse is first described, with the current study, the findings suggest that mito-STAT3 is needed for IL-21 and IL-4 (as described in this paper), but nuclear STAT3 is required for IL-17. It would be great if the authors could better validate this split in STAT3 functions for these cytokines downstream of IL-6. I think this would provide an important distinction in how these cytokines are regulated in the T cells, which is implied by the studies here, but not formally demonstrated*.

We agree with the reviewers that extending our data in vitro to in vivo models of infections could further enhance the quality of the study, and this is something that we consider to follow in the future. Unfortunately, we did not have access to sufficient number of mice for in vivo studies. We were able to do the in vitro experiments suggested by the reviewers with the mut-*Stat3* mice, but we did not have mice for in vivo experiments where a large cohort of mice from the same sex is needed. Neither Dr. O’Shea nor his collaborator at NIH, who also maintains a colony of these mice, could provide us with sufficient mice. As a result, the performance of these studies will require minimum 5-6 months. Thus, we believe these studies are out of the scope of the current study.

Nevertheless, in order to better define the role of Stat3 as a transcription factor versus other functions independent of its transcriptional activity in CD4 cells, we have compared the production of cytokines between *Stat3*^-/-^ and mut-*Stat3* CD4 cells in response to IL-6. In the first version of our manuscript we already showed that IL-21 production in response to IL-6 is totally abrogated in *Stat3-*^*/-*^ CD4 cells, but not in mut-*Stat3* CD4 cells (Figure 6 and Figure 6). We now show that IL-4 production in *Stat3*^*-/-*^ CD4 cells activated with IL-6 is strongly reduced, but is almost not affected in mut-*Stat3* CD4 cells (new Figure 6—figure supplement 2). In contrast, although not affected by IL-6, IL-2 is more reduced in mut-*Stat3* CD4 cells than in *Stat3*^*-/-*^ CD4 cells (new Figure 6—figure supplement 2). Thus, these results further support an alternative role of Stat3 independent of its function as transcription factor that contributes to the production of specific cytokines regulated by IL-6.